# SPICED: A Synaptic Homeostasis-Inspired Framework for Unsupervised Continual EEG Decoding

**Yangxuan Zhou**[1,2], **Sha Zhao**[1,2]*, **Jiquan Wang**[1,2], **Haiteng Jiang**[3,4,1], **Shijian Li**[1,2], **Tao Li**[3,4,1], **Gang Pan**[1,2,4]

[1]State Key Laboratory of Brain-machine Intelligence, Zhejiang University
[2]College of Computer Science and Technology, Zhejiang University
[3]Department of Neurobiology, Affiliated Mental Health Center & Hangzhou Seventh People's Hospital, Zhejiang University School of Medicine
[4]MOE Frontier Science Center for Brain Science and Brain-machine Integration, Zhejiang University
{zyangxuan, szhao, wangjiquan, h.jiang}@zju.edu.cn;
{shijianli, litaozjusc, gpan}@zju.edu.cn;

## Abstract

Human brain achieves dynamic stability-plasticity balance through synaptic homeostasis, a self-regulatory mechanism that stabilizes critical memory traces while preserving optimal learning capacities. Inspired by this biological principle, we propose SPICED: a neuromorphic framework that integrates the synaptic homeostasis mechanism for unsupervised continual EEG decoding, particularly addressing practical scenarios where new individuals with inter-individual variability emerge continually. SPICED comprises a novel synaptic network that enables dynamic expansion during continual adaptation through three bio-inspired neural mechanisms: (1) critical memory reactivation, which mimics brain functional specificity, selectively activates task-relevant memories to facilitate adaptation; (2) synaptic consolidation, which strengthens these reactivated critical memory traces and enhances their replay prioritizations for further adaptations and (3) synaptic renormalization, which are periodically triggered to weaken global memory traces to preserve learning capacities. The interplay within synaptic homeostasis dynamically strengthens task-discriminative memory traces and weakens detrimental memories. By integrating these mechanisms with continual learning system, SPICED preferentially replays task-discriminative memory traces that exhibit strong associations with newly emerging individuals, thereby achieving robust adaptations. Meanwhile, SPICED effectively mitigates catastrophic forgetting by suppressing the replay prioritization of detrimental memories during long-term continual learning. Validated on three EEG datasets, SPICED show its effectiveness. More importantly, SPICED bridges biological neural mechanisms and artificial intelligence through synaptic homeostasis, providing insights into the broader applicability of bio-inspired principles. The source code is available at https://github.com/xiaobaben/SPICED.

---

*Corresponding authors.

# 1   Introduction

During long-term learning and memorization, the human brain dynamically balances information preservation (i.e., stability) and knowledge acquisition (i.e., plasticity) through a self-regulatory mechanism termed **synaptic homeostasis** [1–3]. This homeostasis process coordinates two complementary neural mechanisms: **synaptic consolidation**, which stabilizes critical memory traces, and **synaptic renormalization**, which preserves brain learning capacities [4–10]. The homeostatic interplay between synaptic consolidation and renormalization sustains neural stability-plasticity equilibrium, enabling the brain to dynamically adapt to novel environmental demands and learning tasks. **This biological principle provides critical insights into overcoming stability-plasticity dilemma and catastrophic forgetting in artificial continual learning systems**, especially in non-stationary continual brain decoding scenarios where the pretrained source model is required to adapt to unseen individuals continuously.

For advancing brain decoding research and applications, brain-computer interface (BCI) technology has become instrumental. As the most widely adopted non-invasive BCI technique, electroencephalography (EEG) captures scalp-recorded electrophysiological activity with high temporal resolution, making it particularly valuable in both practical and clinical applications (e.g., sleep staging, depression diagnosis) [11–15]. However, most EEG-based models trained on static datasets exhibit limited generalizability to unseen subjects due to substantial inter-individual variability, hindering their applicability in real-world scenarios where new ones are encountered continually. This problem can be reduced to **continual EEG decoding**, wherein EEG models need to dynamically assimilate new knowledge while retaining consolidated prior representations. Nevertheless, critical challenges persist on continual EEG decoding: *(1) How to enhance model plasticity to enable robust adaptation to each newly emerging individual? (2) How to prevent catastrophic forgetting by suppressing the accumulation of detrimental memory traces during the long-term continual learning?*

Motivated by synaptic homeostasis principles, we leverage the biological synaptic plasticity to develop the neuromorphic framework to address the aforementioned challenges in continual EEG decoding. Here, we propose a novel **S**yna**p**tic Homeostasis-**I**nspired framework for Unsupervised **C**ontinual **E**EG **D**ecoding, called **SPICED**, emulating the brain's mechanisms for processing continuous information streams. The core of SPICED consists of a **biologically inspired synaptic network comprising multiple interconnected synaptic nodes**, with each node storing individual-specific information. Upon encountering a new individual, the synaptic network dynamically incorporates an individual-specific new node and selectively establishes synaptic connections with pre-existing nodes. For challenge *(1)*, we adopt the **synaptic consolidation** mechanism which can automatically and selectively reinforce critical task-discriminative memory traces, to enhance the model plasticity to continuously emerging new individuals. To this end, we need to decide which memory traces are important, and then consolidate them. Specifically, we introduce **critical memory reactivation** mechanism guided by brain's functional specificity to select and replay the important memory trace. During each adaptation, we identify task-discriminative synaptic nodes that are relevant to the individual adaptation, and selectively reactivate them to retrieve memory traces strongly associated with the target individual for replay. We further stabilize these reactivated synaptic nodes by strengthening their connections shown in Fig. 1 (a). As a result, these repeatedly reactivated task-discriminative memory traces undergo progressive consolidation, enhancing their replay prioritization to facilitate

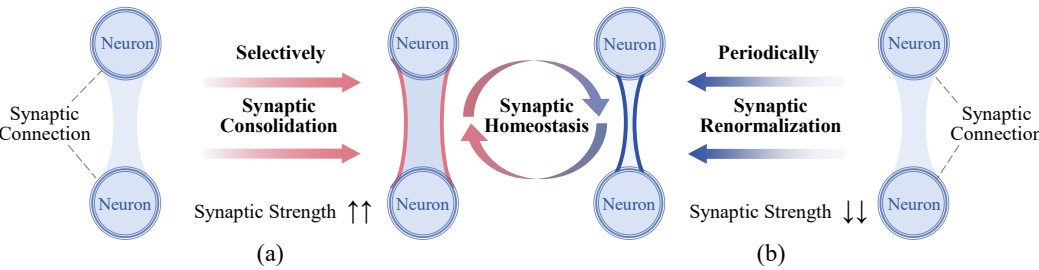

Figure 1: Synaptic consolidation and renormalization mechanisms. (a) Task-driven synaptic consolidation via critical connection strengthening. (b) Stability-preserving synaptic renormalization through connection strength downscaling.

robust adaptation to novel individuals. For challenge *(2)*, we adopt the **synaptic renormalization** mechanism which can maintain a balance of synaptic strength, and restore the ability to learn novel sequences, to suppress the interference from detrimental memory traces. As illustrated in Fig. 1(b), SPICED periodically triggers global downscaling to weaken synaptic connection, preserving the network's learning capacity. The synergistic balance between synaptic consolidation and renormalization in SPICED enables strengthening the task-discriminative memory traces and weakening the redundant noisy memories. By consequently suppressing the replay prioritization of detrimental memories during each individual adaptation, SPICED effectively mitigates catastrophic forgetting caused by error accumulation during long-term continual learning. Our contributions are as follows:

- We propose SPICED, a neuromorphic framework comprising a bio-inspired synaptic network that **integrates the synaptic homeostasis mechanism for unsupervised continual EEG decoding**. SPICED achieves robust adaptation to novel individuals while preventing catastrophic forgetting in long-term learning scenarios.

- We **model the synaptic homeostasis between consolidation and renormalization**, selectively stabilizing task-discriminative memory traces while suppressing interference from redundant noisy memories, achieving balanced network plasticity.

- Validated across three mainstream EEG tasks, SPICED enables continuous dynamic expansion of synaptic networks while maintaining homeostatic balance, thereby achieving robust performance to novel unseen individuals.

## 2 Related Work

### 2.1 Synaptic Homeostasis

Synapses, which bridge pre-synaptic and post-synaptic neurons, function as the essential units for information storage within the brain. Under synaptic homeostasis, their strength and connectivity are dynamically regulated to encode information while preserving circuit stability [1–3, 16]. During active learning states, task-specific neuronal ensembles are selectively recruited through cortical functional specificity, wherein environmental stimuli activate discrete neuronal populations aligned with behavioral demands [17–19]. This recruitment triggers the activity-dependent **long-term potentiation (LTP) driven synaptic consolidation** that selectively strengthens synapses within engaged circuits, thereby stabilizing newly encoded memories[9, 20, 21]. Following learning, during slow-wave sleep (0.5-4 Hz), **long-term depression (LTD) driven synaptic renormalization** globally downscales synaptic strength, preferentially weakening nonessential connections formed during wakefulness[10, 22].

### 2.2 Synaptic-Inspired Continual Learning

Continual learning (CL) aims to enable artificial systems to incrementally acquire knowledge from continuous data streams. Numerous CL methods, such as the regularization-based [23–27], the parameter isolation based [28–30] and the rehearsal-based [31–34] methods, have been developed to mitigate catastrophic forgetting. Recent efforts have integrated synaptic plasticity principles into parameter adaptation and memory consolidation during CL process, demonstrating improved stability-plasticity trade-offs [35–42]. However, these methods only partially emulate neurobiological mechanisms, often necessitating architectural modifications or intrusive adjustments to core learning processes. In contrast, the SPICED framework introduces a novel learning paradigm that biologically emulates synaptic homeostasis through an auxiliary synaptic network, achieving model structural independence while preserving task-specific adaptability and interference suppression capabilities.

### 2.3 Continual EEG Decoding

Numerous deep learning methods have been proposed for EEG decoding across diverse tasks [43–47]. However, parameters of these EEG-based models are typically fixed after training, limiting their generalization ability to new subjects encountered in clinics. This limitation has motivated recent efforts to explore individual-specific continual EEG decoding algorithms in supervised paradigm [48–50]. Zhou et al. [51] introduces an unsupervised individual continual learning (i.e., UICL paradigm) framework tailored for real-world deployment. It incrementally adapts model to unseen individuals

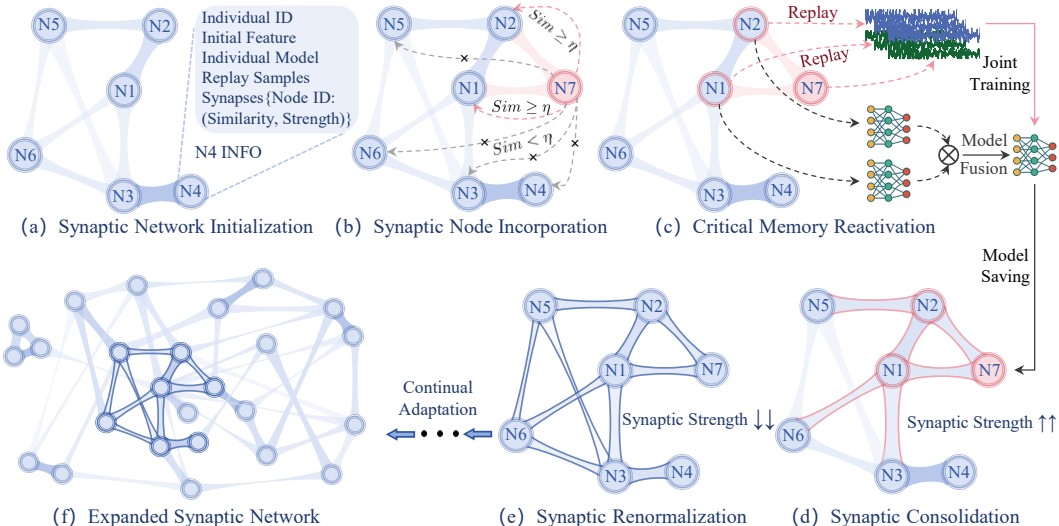

Figure 2: The workflow of SPICED, a synaptic homeostasis-inspired framework for continual EEG decoding. (a) Synaptic network initialization: Constructing individual-specific nodes and initial feature-based synaptic connections. (b-e) Adaptive synaptic dynamics for novel subjects: (b) Synaptic node incorporation: Incremental integration of new individuals; (c) Critical memory reactivation: Replay of critical task-related memories; (d) Synaptic consolidation: Selective strengthening of task-relevant connections; (e) Synaptic renormalization: Global downscaling of synaptic strengths. (f) Expanded network evolution: Expanded synaptic network after continual individual adaptations.

while retaining consolidated knowledge from prior adaptations. Our work is based on the setting of the proposed UICL paradigm.

## 3 Methodology

### 3.1 Overview

To emulate synaptic homeostasis in the human brain, we **consider each subject as a distinct domain** and represent individuals as nodes with one-to-one correspondence to record their unique information (e.g., individual ID, initial feature (i.e., handcrafted features), corresponding model, replayed samples, and synaptic connections). Formally, given multiple labeled individual domains (i.e., source domain) $\mathcal{D}_S = \{\mathcal{X}_S^i, \mathcal{Y}_S^i\}_{i=1}^{\mathcal{N}_S}$ with $\mathcal{N}_S$ subjects, we first utilize the labeled source domain data to pretrain an EEG model and initialize the synaptic network as illustrated in Fig. 2(a). Specifically, we calculate and store each individual's initial feature by extracting features across the time, frequency, and time–frequency domains, with detailed descriptions provided in the Appendix A. We measure inter-nodes initial feature similarity via weighted cosine similarity described as follows:

$$\mathcal{S}(x_i, x_j) = \frac{\omega_{\text{t}} \, \text{sim}(x_i^t, x_j^t) + \omega_{\text{f}} \, \text{sim}(x_i^f, x_j^f) + \omega_{\text{tf}} \, \text{sim}(x_i^{tf}, x_j^{tf})}{\omega_{\text{t}} + \omega_{\text{f}} + \omega_{\text{tf}}} \tag{1}$$

where $x^t$, $x^f$, $x^{tf}$ denote the EEG features vectors in the time, frequency and time-frequency domains, respectively, and $\omega_{\text{t}}$, $\omega_{\text{f}}$, $\omega_{\text{tf}}$ denote their corresponding weights. We define a connection threshold $\xi$, such that synaptic connections are established between nodes whose pairwise similarity exceeds $\xi$, with initial synaptic strength $s_{ij}$ set to 1. Based on inter-nodes similarity, we perform node-wise initialization of the source synaptic network. Then, given multiple unlabeled sequential individual target domains $\mathcal{D}_\mathcal{T} = \{\mathcal{X}_\mathcal{T}^i\}_{i=1}^{\mathcal{N}_\mathcal{T}}$ comprising $\mathcal{N}_\mathcal{T}$ subjects, SPICED continuously adapts to unseen individual target domains by expanding its synaptic network through: (i) incremental node integration, (ii) critical memory reactivation, and (iii) balanced synaptic consolidation-renormalization, as illustrated in Fig. 2(b-e). Following BrainUICL [51], after each adaptation to a new individual, we store high-confidence pseudo-labels and its corresponding model in individual-specific synaptic nodes for further memory replay. Notably, we preserve all labeled samples from source-domain

individuals in their corresponding synaptic nodes. Through continuous integration of new nodes and assimilation of novel knowledge, the synaptic network progressively enhances its robustness shown in Fig. 2(f). Meanwhile, with the interplay of synaptic consolidation and renormalization, critical memories are continuously consolidated while redundant noisy memories are progressively suppressed during continual decoding.

## 3.2 Synaptic Network Initialization

To validate the effectiveness of our SPICED framework, we utilize identical EEG-based model architectures for each downstream EEG task. Specifically, the model integrates a feature extractor for capturing EEG features, a temporal encoder for learning temporal dependencies from EEG sequences, and a classifier adapted to the output requirements of different downstream tasks. Prior to synaptic network initialization, we pre-train the task-specific EEG model on source domains. The detailed model architecture and pre-training process are described in the Appendix B. For each source domain individual, we initialize their synaptic nodes by storing their initial feature, labeled samples, and the pre-trained model. Subsequently, we selectively establish synaptic connections between nodes based on pairwise weighted cosine similarity, and store the synaptic information (e.g., similarity and synaptic strength) for subsequent processing.

## 3.3 Synaptic Node Incorporation

Upon the arrival of a new individual, we first initialize its personal synaptic node (i.e., ID and initial feature). It then traverses pre-existing nodes in the synaptic network to compute pairwise similarity between their initial feature. If the similarity exceeds a predefined connection threshold $\xi$, we establish their inter-nodes connections with an initial strength of 1, and we update the node's synaptic data (e.g., Synapses: {Node1: {similarity: 0.5, strength: 1}}).

## 3.4 Synaptic Consolidation

Human brain leverages its functional specificity to activate task-aligned neuronal ensembles during external stimulation [17, 18]. This neurobiological mechanism ensures cognitive efficiency while enabling adaptive flexibility to address novel environmental demands. Inspired by this mechanism, we introduce the **Critical Memory Reactivation** to activate task-relevant memory traces into ongoing individual-specific adaptation. To determine which nodes to reactivate, we **introduce a synaptic importance coefficient to measure connection significance** between nodes $\mathcal{N}_i$ and $\mathcal{N}_j$ as follows:

$$\mathcal{I}(\mathcal{N}_i, \mathcal{N}_j) = \alpha \mathcal{S}(x_i, x_j) + (1 - \alpha)\bar{s}_j; \quad \bar{s}_j = \frac{1}{\mathcal{N}} \sum_{k=1}^{\mathcal{N}} s_{jk} \quad (2)$$

Here, $\mathcal{I}(\mathcal{N}_i, \mathcal{N}_j)$ denotes the importance of node $\mathcal{N}_j$ relative to node $\mathcal{N}_i$. $\mathcal{S}(x_i, x_j)$ denotes the similarity between $\mathcal{N}_i$ and $\mathcal{N}_j$. $\bar{s}_j$ denotes the average synaptic connection strength of node $\mathcal{N}_j$, reflecting the global activation status of its associated neural pathways within the synaptic network. And $\alpha$ denotes the importance weight of inter-nodes similarity.

### 3.4.1 Critical Data Replay

We calculate and rank the importance coefficient of synaptic connections for the current node in descending order. To prioritize the activation of the most relevant memories, we select top-K node samples via importance-weighted sampling for weighted replay.

### 3.4.2 Critical Model Fusion

In traditional continual learning paradigms, the incremental model $\mathcal{M}_i$ at the $i$-th step is derived from training on the prior model $\mathcal{M}_{i-1}$, exclusively dependent on knowledge consolidated during the terminal phase of prior training cycles. In contrast, **the human brain selectively recruits task-aligned neuronal assemblies during cognitive processing, integrating information streams from all historical experiences without dependence on a single network state**. This biological insight motivates the integration of synaptic networks to determine optimal activation strategies. Specifically, for the current $i$-th individual, we select the top-K synaptic nodes based on importance coefficient

ranking. Then we initialize the fusion model $\mathcal{M}_i$ through weighted aggregation of corresponding top-K node model parameters, as detailed below:

$$\mathcal{M}_i = \sum_{j=1}^{K} \frac{\mathcal{I}(\mathcal{N}_i, \mathcal{N}_j)}{\sum_{k=1}^{K} \mathcal{I}(\mathcal{N}_i, \mathcal{N}_k)} \mathcal{M}_j \tag{3}$$

Following the BrainUICL, we clone a guidance model from the fused model $\mathcal{M}_i$ and conduct self-supervised training on newly added individual samples using this guidance model, as detailed in the Appendix C. The guidance model generates high-quality pseudo-labels with predicted probabilities surpassing a predefined threshold $\eta$. Then, pseudo-labeled samples undergo joint training with incremental samples via cross-entropy loss, as formulated below:

$$\mathcal{L} = -(\beta \sum_c \tilde{y}_c \log \hat{y}_c + (1 - \beta) \sum_c y_{\mathcal{R}_c} \log \hat{y}_{\mathcal{R}_c}) \tag{4}$$

where $\beta$ is a hyper-parameter controlling the weight of the loss function, while $y_c$ and $y_{\mathcal{R}_c}$ represent the pseudo-labels of incremental individual samples and the ground-truth labels of replayed top-K node's samples, respectively. Following joint training, the fusion model $\mathcal{M}_i$ is used to predict incremental samples, and generates high-confidence pseudo-labeled samples. Both the pseudo-labeled samples and the fine-tuned model $\mathcal{M}_i$ are stored in node $\mathcal{N}_i$ to for future memory replay. **Notably, if no synaptic connections exist for a new node**, during the model fusion phase, we perform fusion using the three most similar node models based on inter-nodes similarity. The fused model is then directly applied to self-supervised learning for adaptation without employing replay strategies.

### 3.4.3 Critical Connection Consolidation

Upon completion of incremental individual adaptation, the SPICED conducts synaptic consolidation to simulate activity-dependent long-term potentiation (LTP) mechanisms, stabilizing the task-relevant node memories reactivated in critical memory reactivation. Specifically, we only strengthen the synaptic strengths of the top-K activated nodes, as described below:

$$s'_{ij} = \gamma s_{ij}, \quad \forall i \in \mathcal{T}_K, \ \forall j \in \mathcal{N}_i, \tag{5}$$

Here, $i$ indexes the top-K activated nodes, $j$ indexes all nodes connected to node $\mathcal{N}_i$ and $\gamma$ represents the consolidation coefficient governing synaptic strength amplification. In biological neural systems, **synaptic strength does not increase indefinitely with repeated activation due to metabolic homeostasis constraints**. Studies indicate that synaptic efficacy typically saturates at 150–300% of baseline levels [52–54]. In SPICED, we impose an upper limit of 3 on synaptic strength (with a baseline strength of 1) to align computational efficiency with biological plausibility.

---

**Algorithm 1:** SPICED framework for Unsupervised Individual Continual Learning

---

**Input:** $\{\mathcal{X}_{\mathcal{S}}^i, \mathcal{Y}_{\mathcal{S}}^i\}_{i=1}^{\mathcal{N}_{\mathcal{S}}}$, $\{\mathcal{X}_{\mathcal{T}}^i\}_{i=1}^{\mathcal{N}_{\mathcal{T}}}$
**Output:** Synaptic Network $\mathcal{S}_{\mathcal{N}}$
**Synaptic Network Initialization:**
Pretrain the source model $\mathcal{M}_0$ using $\{\mathcal{X}_{\mathcal{S}}^i, \mathcal{Y}_{\mathcal{S}}^i\}$
Initialize the $\mathcal{S}_{\mathcal{N}}$ using $\{\mathcal{X}_{\mathcal{S}}^i, \mathcal{Y}_{\mathcal{S}}^i\}$ and $\mathcal{M}_0$, guided by inter-individual similarity Eq. (1).
**Unsupervised Individual Continual Learning:**
**for** $i \leftarrow 1$ *to* $\mathcal{N}_{\mathcal{T}}$ **do**
  Initialize the synaptic node $\mathcal{N}_i$ for incremental individual and incorporate it into $\mathcal{S}_{\mathcal{N}}$.
  Compute the top-K most critical nodes for all nodes relative to $\mathcal{N}_i$ according to Eq. (2).
  Weightedly replay samples from the top-K nodes.
  Aggregate the top-K models via weighted fusion to obtain model $\mathcal{M}_i$ by Eq. (3).
  Generate the guiding model $\mathcal{M}_g$, copied from the model $\mathcal{M}_i$;
  Optimize $\mathcal{M}_g$ by minimizing Eq. (7) and generate confident pseudo labels
  Optimize $\mathcal{M}_i$ by minimizing Eq. (4).
  Incorporate the high-confidence pseudo-labeled samples $(\tilde{\mathcal{X}}_{\mathcal{T}}^i, \tilde{\mathcal{Y}}_{\mathcal{T}}^i)$ and model $\mathcal{M}_i$ into $\mathcal{N}_i$.
  Strengthen the synaptic strength of top-K activated nodes according to Eq. (5).
  Weaken the synaptic strength of global nodes according to Eq. (6).
**end**

---

## 3.5 Synaptic Renormalization

The proposed SPICED periodically triggers synaptic renormalization to simulate sleep-dependent long-term depression (LTD) mechanisms, globally weakening synaptic connection strengths. Inspired by the Ebbinghaus forgetting curve [55], which models the temporal decay of human memory retention, we formalize node-specific dynamic renormalization rates to regulate synaptic strength in a time-dependent manner, as detailed below:

$$s''_{ij} = e^{-\frac{t_i}{\lambda}} \cdot s'_{ij}, \quad \forall i \in \mathcal{V}, \ \forall j \in \mathcal{N}_i, \tag{6}$$

where $\mathcal{V}$ denotes the set of all synaptic nodes, $j$ indexes all nodes connected to node $\mathcal{N}_i$ and $\lambda$ is the decay factor. Notably, upon initialization of the $i$-th new node, the time step $t_i$ is set to 1. During global synaptic renormalization Eq. (6), $t_i$ is incremented by 1 for all nodes. However, when the current node is activated and undergoes synaptic consolidation Eq. (5), $t_i$ is reset to 1. This mechanism ensures that **infrequently activated nodes exhibit accelerated forgetting (higher decay rates), while frequently stabilized nodes maintain lower forgetting rates, aligning with neurobiological learning principles**. The homeostasis interplay between synaptic consolidation and renormalization effectively consolidates critical information and enhances model adaptability to novel individuals. Meanwhile, it suppresses the redundant information, thereby preventing catastrophic forgetting caused by error accumulation. The overall algorithm is illustrated in Algorithm 1.

# 4 Experiment

## 4.1 Experimental Setup

Table 1: Overview of downstream BCI tasks and datasets.

| BCI Tasks | Datasets | # Subjects | Rate | # Channels | Duration | # Samples | Label |
|---|---|---|---|---|---|---|---|
| I. Sleep Staging | ISRUC | 100 | 200Hz | 8 | 30s | 89,240 | 5-class |
| II. Emotion Recognition | FACED | 123 | 250Hz | 32 | 10s | 10,332 | 9-class |
| III. Motor Imagery | Physionet | 109 | 160Hz | 64 | 4s | 9,837 | 4-class |

We selected three mainstream EEG-based task datasets for validation: ISRUC [56], FACED [57] and Physionet-MI [58] shown in Tab. 1. Each dataset comprises at least 100 subjects, enabling effective evaluation of the SPICED framework's performance under long-term UICL scenarios. Detailed dataset descriptions and preprocessing procedures are provided in Appendix D, while more detailed experimental settings, including hyper-parameters and CL training details, are listed in Appendix E.

**Each dataset is partitioned into a pretraining set (i.e., source domain)** for pre-training the source model and **an incremental set (i.e., target domain)** for evaluating the performance of the SPICED framework in unsupervised individual continual learning (i.e., continual EEG decoding) scenario. For each novel individual in the incremental set, model performance is evaluated using **Accuracy (ACC)** and **Macro-F1 (MF1)** metrics both before and after individual domain adaptation, as defined follows:

$$\text{Accuracy} = \frac{1}{N} \sum_{i=1}^{N} \mathbf{1}(y_i = \hat{y}_i) \qquad \text{Macro-F1} = \frac{1}{C} \sum_{c=1}^{C} \frac{2 \cdot \text{Precision}_c \cdot \text{Recall}_c}{\text{Precision}_c + \text{Recall}_c}$$

Notably, for the $i$-th incremental individual, the SPICED framework initializes model $\mathcal{M}_i$ by **integrating information across the holistic synaptic network**, rather than relying on the **last incremental model $\mathcal{M}_{i-1}$ from preceding temporal states**.

## 4.2 Results and Analysis

### 4.2.1 Overall Performance

We have evaluated the SPICED framework on three different downstream EEG datasets. Specifically, we set varying source-target (i.e., pretrain-incremental) dataset splits (source proportions: 10%-50%) to evaluate the long-term continual EEG decoding performance of the SPICED framework under few-shot pre-training conditions. Notably, the input order of incremental individuals directly

Table 2: Overview performance of SPICED on three downstream EEG tasks under different source-target individuals splits (e.g., "10%" denotes that 10% of subjects are used to pretrain the source model, with the remaining 90% serving as incremental individuals). Notably, distinct synaptic connection thresholds $\xi$ were configured for the three datasets, while consistency was maintained across all other hyper-parameters.

| | ISRUC ($\xi = 0.1$) | | | | FACED ($\xi = 0.4$) | | | | Physionet-MI ($\xi = 0.5$) | | | |
| --- | --- | --- | --- | --- | --- | --- | --- | --- | --- | --- | --- | --- |
| | Average ACC | | Average MF1 | | Average ACC | | Average MF1 | | Average ACC | | Average MF1 | |
| | $\mathcal{M}_0$ | $\mathcal{M}_i$ | $\mathcal{M}_0$ | $\mathcal{M}_i$ | $\mathcal{M}_0$ | $\mathcal{M}_i$ | $\mathcal{M}_0$ | $\mathcal{M}_i$ | $\mathcal{M}_0$ | $\mathcal{M}_i$ | $\mathcal{M}_0$ | $\mathcal{M}_i$ |
| 10% | 56.8 | 62.6±1.35 (**5.4**↑) | 48.4 | 55.4±1.51 (**7.0**↑) | 23.5 | 25.7±0.10 (**2.2**↑) | 19.5 | 22.7±0.17 (**3.2**↑) | 40.6 | 43.4±0.44 (**2.8**↑) | 36.2 | 42.0±0.46 (**5.8**↑) |
| 20% | 57.3 | 69.2±0.45 (**11.9**↑) | 44.1 | 61.7±0.66 (**17.6**↑) | 26.8 | 32.8±0.77 (**6.0**↑) | 23.8 | 30.8±1.12 (**7.0**↑) | 42.0 | 45.3±0.20 (**3.3**↑) | 38.8 | 43.8±0.21 (**5.0**↑) |
| 30% | 66.8 | 75.6±0.13 (**8.8**↑) | 60.5 | 71.4±0.07 (**10.9**↑) | 31.7 | 43.7±0.51 (**12.0**↑) | 27.2 | 41.7±0.58 (**14.5**↑) | 42.2 | 48.7±0.17 (**6.5**↑) | 37.9 | 47.9±0.24 (**10.0**↑) |
| 40% | 73.5 | 77.8±0.35 (**4.3**↑) | 68.6 | 72.3±0.43 (**3.7**↑) | 38.4 | 48.4±0.32 (**10.0**↑) | 35.5 | 46.2±0.46 (**10.7**↑) | 47.7 | 50.7±0.14 (**3.0**↑) | 44.8 | 49.6±0.13 (**4.8**↑) |
| 50% | 65.0 | 74.5±0.39 (**9.5**↑) | 58.5 | 69.2±0.37 (**10.7**↑) | 43.4 | 48.9±0.42 (**5.4**↑) | 41.7 | 47.0±0.50 (**5.3**↑) | 47.9 | 52.4±0.11 (**4.5**↑) | 44.5 | 51.2±0.14 (**6.7**↑) |

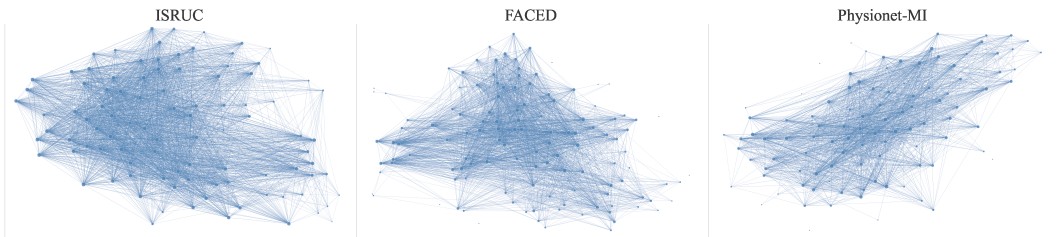

Figure 3: The visualization of the expanded synaptic network across ISRUC, FACED and Physionet-MI datasets, with connection thresholds $\xi$ set to 0.1, 0.4, and 0.5, respectively. Notably, each point corresponds to an individual synaptic node, while dashed lines denote synaptic connections between nodes. Node size and dashed line thickness are proportional to the number of synaptic connections associated with each node.

influences synaptic network expansion trajectories during long-term continual decoding. Therefore, for each source-target domain split ratio, we **maintained consistent dataset partitioning** while **altering the input order** of incremental individuals across five repetitions, so as to **evaluate the statistical reliability of experimental results**. For incremental individuals, we report their personal performance of the initial source model $\mathcal{M}_0$ and the domain-adapted individual model $\mathcal{M}_i$, as detailed in Tab. 2. Experimental results demonstrate that the SPICED framework maintains robust performance in long-term unsupervised individual continuous EEG decoding scenarios under few-shot supervised pre-training conditions. Furthermore, the SPICED framework demonstrates consistent stability across varying input order of incremental individuals, reflected in low standard deviations of accuracy and macro-F1 metrics. Besides, it is worth noting that for resting-state sleep datasets such as ISRUC, significant inter-individual variability in initial features necessitates a lower connection threshold $\xi$ to enhance synaptic connection density. Conversely, for the task-based datasets such as FACED and PhysioNet-MI, the individual initial features exhibit lower variability under identical task stimuli. Therefore, we set $\xi$ much higher to filter redundant homogeneous synaptic connections while preserving individual-discriminative connections. We employed t-SNE [59] to visualize the expanded synaptic networks across three datasets shown in Fig. 3. Central nodes (i.e., larger-sized ones) exhibit dense synaptic connections with high strength (i.e., thicker dashed lines), suggesting **frequent activation and consolidation processes** during continual learning to facilitate adaptation to new individuals. In contrast, peripheral nodes (i.e., smaller-sized ones) exhibit sparse or absent synaptic connections, reflecting deviations of their initial features from dataset-level characteristics, thus being classified as outliers. Synaptic renormalization **prioritized weakening** these connections to mitigate noise introduced during critical memory reactivation, thereby enhancing training stability.

### 4.2.2 Comparison with Other Methods

We have compared our method with other existing methods at different source domain proportions: **MMD [60]**: a UDA method that minimizes the Maximum Mean Discrepancy between feature distributions. **EWC [23]**: a classical regularization-based CL method. **UCL-GV [61]**: a contrastive alignment method for continual domain adaptation. **ReSNT [48]**: a dynamic memory evolution based method for continual EEG decoding. **CoUDA [62]**: a recent method for continual domain

Table 3: Accuracy performance comparison with other methods at different source domain proportions. Each method was evaluated under the same dataset partitioning, with five runs conducted per method by randomly shuffling the input order of the incremental set to enable statistical evaluation.

|  | ISRUC | | | FACED | | | Physionet-MI | | |
|---|---|---|---|---|---|---|---|---|---|
|  | 10% | 30% | 50% | 10% | 30% | 50% | 10% | 30% | 50% |
| MMD | 57.0±0.47 | 70.2±0.83 | 68.2±1.00 | 24.1±0.24 | 39.0±1.48 | 19.1±2.59 | 37.3±0.34 | 44.1±0.38 | 47.9±0.54 |
| EWC | 60.6±0.71 | 71.3±0.49 | 72.5±0.39 | 22.3±0.36 | 39.5±0.94 | 48.7±0.66 | 42.8±1.19 | 47.3±0.58 | 50.2±0.43 |
| UCL-GV | 52.5±1.16 | 72.5±0.32 | 73.9±0.29 | 25.0±0.43 | 41.4±1.50 | 22.8±3.64 | 33.1±0.35 | 39.7±0.22 | 45.5±0.71 |
| ReSNT | 56.4±0.70 | 71.3±0.89 | 72.1±0.69 | 14.1±1.66 | 33.3±8.10 | 23.6±3.29 | 35.9±1.49 | 44.1±0.50 | 48.8±0.28 |
| CoUDA | 52.7±0.44 | 73.3±0.23 | 72.3±0.94 | 23.9±0.16 | 39.2±1.06 | 22.5±2.61 | 41.6±0.25 | 44.2±0.31 | 51.4±0.16 |
| BrainUICL | 56.9±0.46 | 74.8±0.11 | 73.6±0.29 | 21.2±2.04 | 33.8±2.90 | 20.2±3.46 | 43.1±0.33 | 48.3±0.38 | **52.5±0.36** |
| SPICED | **62.6±1.35** | **75.6±0.13** | **74.5±0.39** | **25.7±0.10** | **43.7±0.51** | **48.9±0.42** | **43.4±0.44** | **48.7±0.17** | 52.4±0.11 |

adaptation. **BrainUICL [51]**: a dynamic framework designed for unsupervised continual EEG decoding. We implemented these methods based on our setting. Notably, each method was evaluated under the same dataset partitioning, with five repeats conducted per method by randomly shuffling the input order of the incremental set to enable statistical evaluation. The experimental results indicate that our SPICED method not only outperforms existing approaches but also exhibits superior robustness. This is attributed to SPICED's mechanism of leveraging critical memory across the holistic synaptic network for individual adaptation, rather than exclusively depending on the latest model $\mathcal{M}_{i-1}$ as in conventional continual domain adaptation paradigms. Such strategy substantially alleviates the disruptive effects of outlier shifts within the continual learning trajectory on the current individual. For instance, while BrainUICL achieves performance comparable to SPICED in certain experimental settings, it exhibits significantly degraded performance in several scenarios (e.g., 10% source proportion on ISRUC, 50% source proportion on FACED).

### 4.2.3 Interplay Analysis of Varying Hyper-parameters $\lambda$ and $\gamma$

The interplay between synaptic consolidation and renormalization collectively facilitates synaptic homeostasis. In this section, we conduct hyper-parameter studies on the synaptic renormalization decay factor $\lambda$ and consolidation coefficient $\gamma$ on the ISRUC dataset, as depicted in Fig. 4. The left sub-figure visualizes the variations in synaptic renormalization rate across distinct $\lambda$ values. As $\lambda$ increases, the rate of decline in renormalization diminishes progressively with step increments. The experimental results demonstrate that the SPICED framework achieves optimal performance when $\lambda$ is set to 30 and $\gamma$ to 1.3. This parameterization ensures a stabilized synaptic renormalization decay profile and moderate consolidation. This balance prevents synaptic strength collapse from transient suppression or explosion from recurrent activation.

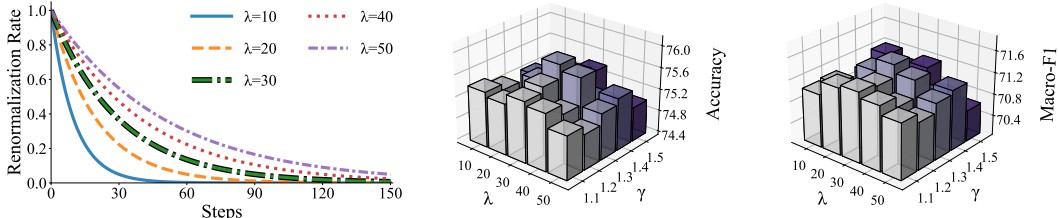

Figure 4: Analysis of synaptic renormalization decay factor $\lambda$ and consolidation coefficient $\gamma$ on the ISRUC dataset with 30% source domain proportion. The left sub-figure illustrates the variation in synaptic renormalization rate across different decay factor $\lambda$ values.

### 4.2.4 Dynamics of the Average Synaptic Strength

In this section, we visualize the dynamics of average synaptic connection strength in source domain nodes across three datasets during unsupervised continual domain adaptation, with all nodes initialized to an average synaptic strength of 1. As shown in Fig. 5, experimental results demonstrate that as the synaptic network expands, the dynamics of average synaptic strength changes exhibit a

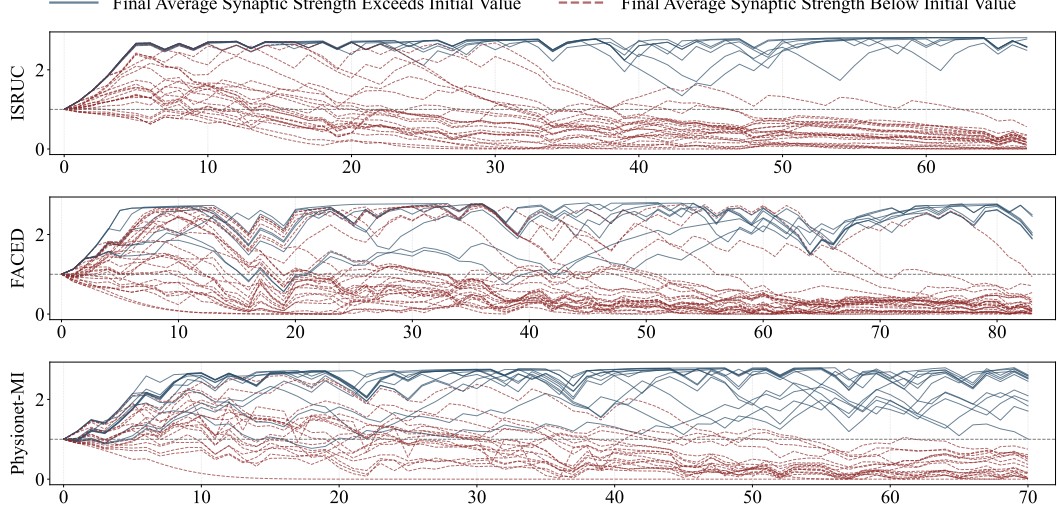

Figure 5: Dynamics of average synaptic connection strength in source domain nodes during synaptic network expansion: with color-coded trajectories indicating post-expansion potentiation or depression compared to initial strength.

consistent pattern across three datasets. Critical nodes undergo persistent strengthening of their synaptic strength (i.e., blue lines), thereby enhancing their influence in subsequent learning tasks such as adaptation to new individuals. This persistent consolidation mechanism mimics neural LTP, where short-term memory representations are gradually consolidated from the hippocampus into cortical long-term engrams. Conversely, redundant connections undergo progressive weakening through synaptic renormalization (i.e., red lines), mitigating noise-induced disruptions during domain adaptation. This process mirrors the LTD-driven passive forgetting mechanism in the brain, which dynamically reallocates cognitive resources by suppressing task-irrelevant or low-salience information. In summary, our synaptic homeostasis-inspired SPICED framework enables long-term continual learning by consolidating task-discriminative features and suppressing redundant noise. It synergizes with the subsequent functionally specialized critical memory activation module to jointly enable robust unsupervised continual EEG decoding.

More detailed visualization experiments are provided in the Appendix F, H. And more detailed additional studies are listed in Appendix G, I, J, K.

## 5 Conclusion

In this work, we propose SPICED—a neuromorphic framework integrating synaptic homeostasis principles for unsupervised continual EEG decoding in non-stationary scenarios. SPICED comprises a novel synaptic network that enables dynamic expansion during continual adaptation via three biologically inspired mechanisms. First, we introduced critical memory reactivation to selectively activate task-relevant memory for replay during each adaptation. Second, we adopt the synaptic consolidation to strengthen these critical memory traces. Third, we adopt the synaptic renormalization to periodically weaken inter-node connection strengths globally. The synergistic balance between these bio-inspired mechanisms enables robust adaptation to each newly emerging individual and prevention of catastrophic forgetting during long-term continual EEG decoding. SPICED achieves robust performance across three downstream EEG tasks. More discussion about implications and limitations is provided in Appendix L.

## 6 Acknowledgments

This work was supported by STI 2030 Major Projects (2021ZD0200400), the National Natural Science Foundation of China (No.62476240) and the Key Program of the Natural Science Foundation of Zhejiang Province, China (No. LZ24F020004). The corresponding author is Dr. Sha Zhao.

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

## A  Initial Feature Extraction

In this work, the synaptic network construction is driven by inter-individual similarity matching of initial features, with synaptic connections formed when pairwise initial feature similarity exceeds a predefined threshold $\xi$. Thus, the extraction of individual initial feature constitutes a critical prerequisite for the framework. We extract individual-specific initial EEG features from raw signals via **channel-wise feature engineering** across three dimensions—time domain, frequency domain, and time-frequency domain. **Time Domain Feature:** For each EEG channel, we extract the mean, variance, kurtosis, skewness, and Hjorth parameters (mobility and complexity) as initial time-domain features. **Frequency Domain Feature:** For each EEG channel, we extract the power spectral density (PSD) across five frequency bands (i.e., $\delta$ (0.5-4Hz), $\theta$ (4-8Hz), $\alpha$ (8-13)Hz, $\beta$ (13-30Hz), $\gamma$ (30-45Hz)) as initial frequency-domain features. **Time-Frequency Domain Feature:** For each EEG channel, we extract wavelet energy features as initial time-frequency domain features. Following channel-wise feature extraction, inter-channel normalization is applied to the initial feature vectors to mitigate scaling discrepancies caused by dimensional heterogeneity across features.

## B  More Details for Experimental Settings on Pre-training

To validate the effectiveness of the SPICED framework across diverse downstream EEG tasks, we employ a uniform model architecture comprising three core components: (1) a feature extractor composed of multiple layers of CNN, (2) a Transformer-based encoder for contextual representation learning, and (3) a task-specific classifier composed of multiple fully connected layers. The detailed model parameters and pre-trained settings are summarized in Table. 4.

Table 4: Hyper-parameters of the pre-trained model and detailed pre-training configurations. For the Conv1D layer, the parameter from left to right corresponds to: filters , kernel size , and stride.

| | | |
|---|---|---|
| | Epoch | 100 |
| | Learning Rate | 1e-4 |
| | Optimizer | AdamW |
| | Adam $\beta$ | (0.5, 0.99) |
| Pre-training | Weight Decay | 3e-4 |
| | Batch | 32 |
| | Dropout | 0.1 |
| | Clipping gradient norm | 1 |
| | 1-th Conv1D | (64, 50, 6) |
| | 1-th MaxPool1D | (8, 8) |
| | 2-th Conv1D | (128, 8) |
| CNN Blocks | 3-th Conv1D | (256, 8) |
| | 4-th Conv1D | (512, 8) |
| | 4-th MaxPool1D | (4, 4) |
| | Layers | 3 |
| | Hidden Dimension | 512 |
| Transformer | Heads | 8 |
| | Feed-forward dimension | 2048 |
| | 1-th Linear Layer | (encoder output, 256) |
| Classifier | 2-th Linear Layer | (256, 128) |
| | 3-th Linear Layer | (128, task-specific output) |

## C  More Details for Self-Supervised Learning

Following the BrainUICL framework, we apply a self-supervised learning (SSL) method on newly incremental individual to fine-tune the guided model cloned from the initial fusion model $\mathcal{M}_i$, so as to generate high-confidence pseudo-labels for subsequent joint-training. Specifically, we employ the Contrastive Predictive Coding (CPC) [63] algorithm to adapt the feature extractor and encoder to the data distribution of new individuals by predicting future EEG sequences using prior contextual information. Formally, given a latent EEG sequences $H = \{h_0, h_1, h_2, h_3, ..., h_t, h_{t+1}, h_{t+2}, h_{t+3}\}$ encoded by the feature encoder, we instantiate a Transformer encoder as an autoregressive model to encode the early-stage sequential representation $H_{i \leq t}$ into a contextual vector $c_t$. Subsequently,

multiple linear layers are initialized to predict future EEG representations $\{h_{t+1}, h_{t+2}, h_{t+3}\}$ conditioned on $c_t$, formulated as $z_{t+k} = f_k(c_t)$ for k=1, 2, 3, where $z_{t+k}$ denotes the predicted time steps for $h_{t+k}$. Then we employ the contrastive loss to update the feature extractor and encoder as follows:

$$\mathcal{L}_{\text{cpc}} = -\underset{H_b}{\mathbb{E}}[log\frac{exp(h_{t+k}^T(f_k(c_t)))}{\sum_{h_j \in H_b} exp((h_j^T f_k(c_t)))}] \tag{7}$$

## D    More Details for Dataset Preparation

**ISRUC** is a publicly sleep dataset comprising three sub-groups. In this study, we employ sub-group1 of the ISRUC dataset, comprising 100 all-night polysomnography (PSG) recordings from adult subjects. The EEG signals were acquired using six channels (F3-A2, C3-A2, O1-A2, F4-A1, C4-A1, and O2-A1) with a sampling frequency of 200 Hz. Additionally, two electrooculography (EOG) channels (E1-M2 and E2-M1) were included as input modalities. All EEG and EOG signals are were bandpass filtered (0.3 Hz–45 Hz) and resampled to 100 Hz. The signals are segmented into 30-second epochs, which are manually annotated by sleep experts into five sleep stages (Wake, N1, N2, N3, and REM) in accordance with the American Academy of Sleep Medicine (AASM) [64] guidelines. Following previous sleep studies [65], we formulate the sleep staging task as a sequence-to-sequence classification problem, with a sequence length of 20 corresponding to 10-minute sleep epochs ($20 \times$ 30-second epochs). Notably, subjects 8 and 40 were excluded due to incomplete channels.

**FACED** is a large-scale, high-resolution affective computing dataset that encompasses nine distinct emotional states: amusement, inspiration, joy, tenderness, anger, fear, disgust, sadness, and neutral. It comprises recordings from 123 participants, with each session captured via a 32-channel EEG system at a sampling frequency of 250 Hz. All signals were segmented into 10-second epochs for analysis, and the full cohort of 123 recordings was utilized for experimental evaluation.

**Physionet-MI** is a motor imagery EEG dataset encompassing four distinct movement categories—left fist, right fist, both fists, and both feet—derived from 109 participant recordings. Each recording contains 64-channel EEG signals with a sampling frequency of 160 Hz. All recordings were partitioned into 4-second epochs for analysis. Notably, subjects 38, 88, 89, 92, 100 and 104 were excluded due to incomplete channels.

## E    More Details for Experimental Settings on Continual Learning

The detailed continual learning configurations and synaptic network hyper-parameters are summarized in Table 5.

Table 5: Detailed continual learning configurations and hyper-parameters of the synaptic network.

| | | |
|---|---|---|
| | SSL Epoch | 10 |
| | CL Epoch | 10 |
| | SSL Learning Rate | 1e-7 |
| | CL Learning Rate | 1e-7 |
| Continual Learning | Optimizer | AdamW |
| | Adam $\beta$ | (0.5, 0.99) |
| | Weight Decay | 3e-4 |
| | Batch | 32 |
| | Dropout | 0.1 |
| | ISRUC Connection Threshold $\xi$ | 0.1 |
| | FACED Connection Threshold $\xi$ | 0.4 |
| | Physionet-MI Connection Threshold $\xi$ | 0.5 |
| | Time Domain Weight $\omega_t$ | 0.9 |
| | Frequency Domain Weight $\omega_f$ | 1.5 |
| Synaptic Network | Time-Frequency Domain Weight $\omega_{tf}$ | 1.2 |
| | Importance Weight $\alpha$ | 0.2 |
| | Top-K | 15 |
| | Pseudo Label confidence threshold $\eta$ | 0.9 |
| | Loss Function Weight $\beta$ | 0.7 |
| | Renormalization Decay Factor $\lambda$ | 30 |
| | Consolidation Factor $\gamma$ | 1.3 |

# F   Synaptic Network Visualization under Varying Connection Thresholds

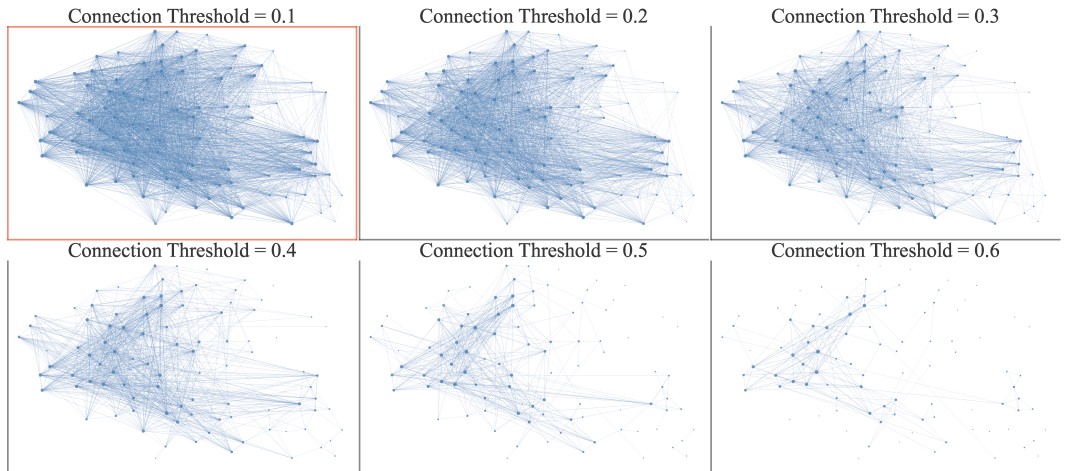

Figure 6: Visualization of the expanded synaptic network in the ISRUC dataset.

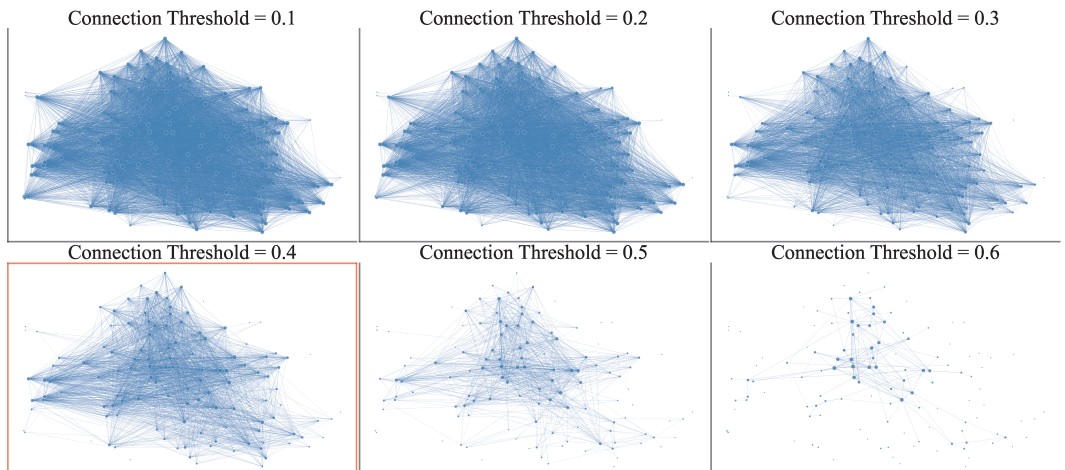

Figure 7: Visualization of the expanded synaptic network in the FACED dataset.

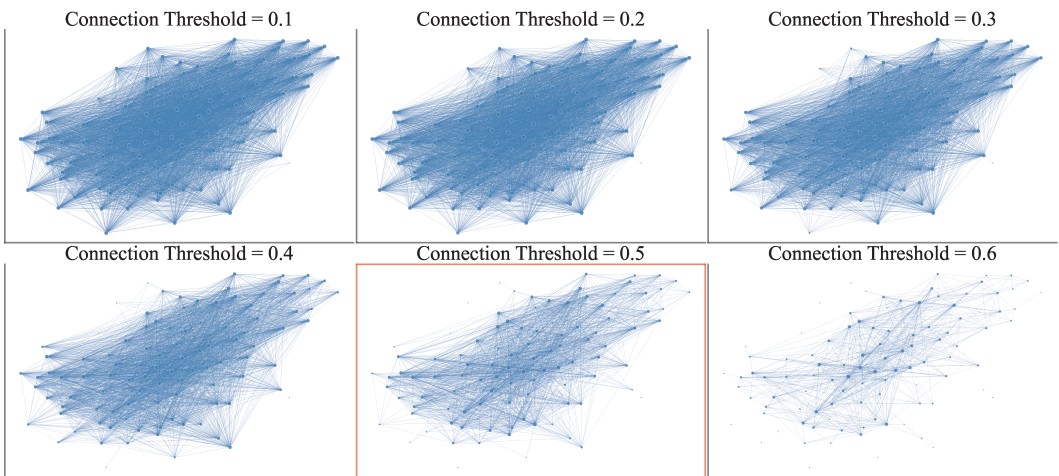

Figure 8: Visualization of the expanded synaptic network in the Physionet-MI dataset.

Table 6: Overview of varying connection threshold $\xi$ experiment performance with 30% source domain proportion.

| $\xi$ | ISRUC | | FACED | | Physionet-MI | |
|---|---|---|---|---|---|---|
| | Average ACC | Average MF1 | Average ACC | Average MF1 | Average ACC | Average MF1 |
| 0.1 | **75.5** | **71.1** | 43.7 | 41.8 | 48.7 | 47.86 |
| 0.2 | 74.9 | 70.7 | 43.9 | 42.0 | 48.8 | 47.92 |
| 0.3 | 75.0 | 71.0 | 43.6 | 41.5 | 48.6 | 47.75 |
| 0.4 | 74.1 | 70.0 | **44.2** | **42.3** | 48.6 | 47.71 |
| 0.5 | 73.4 | 69.4 | 43.7 | 41.5 | **48.9** | **48.11** |
| 0.6 | 73.9 | 69.7 | 43.5 | 41.4 | 48.7 | 47.77 |

We have visualized the synaptic networks derived from three datasets under varying connection thresholds $\xi$, as depicted in the Fig. 6, Fig. 7 and Fig. 8. In the ISRUC resting-state sleep dataset, significant inter-individual variability in initial features causes a sharp reduction in synaptic connections as the connection threshold increases. To preserve synaptic connection diversity, we configured a lower threshold (i.e., $\xi = 0.1$ ). In contrast, for task-based datasets such as FACED and Physionet-MI, identical stimuli elicit initial features that result in an overabundance of redundant synaptic connections at low thresholds (i.e., $\xi \leq 0.3$).Consequently, dataset-specific thresholds were established to filter low-quality synaptic connections. Detailed experimental results for the connection thresholds are summarized in Tab. 6. The results demonstrate that the SPICED framework achieves optimal performance across all datasets under these dataset-specific thresholds. Notably, visualization analyses reveal that synaptic network connection densities converge to a consistent level (i.e., the subplots highlighted by red boxes) when the optimal thresholds are applied for each dataset.

## G   Hyper-parameter Analysis of Importance Weight $\alpha$

In this section, we investigate the importance weight $\alpha$ to quantify the influence of inter-node similarity and average synaptic connection strength on node importance, as formulas in Eq. (2). As shown in Fig. 9, the experimental results demonstrate that SPICED achieves superior performance across datasets with smaller $\alpha$ values (where inter-node similarity receives lower weighting) compared to scenarios with larger $\alpha$ values (where average synaptic connection strength is assigned reduced weights). To sum op, the average synaptic connection strength of target nodes exerts a greater influence on node importance compared to inter-node similarity, as evidenced by the following reasons. First, inter-node similarity is derived from individual initial features and remains unaffected by subsequent learning processes, thereby lacking the flexibility to dynamically assess node importance. Second, the average synaptic connection strength dynamically evolves through synaptic consolidation and renormalization, enabling a more accurate reflection of a node's significance within the synaptic network. A higher average synaptic connection strength signifies greater generalizability of node samples, and activating such nodes enhances the model's adaptability to new individuals. Based on the aforementioned analysis, we uniformly set $\alpha$ to 0.2 for all datasets.

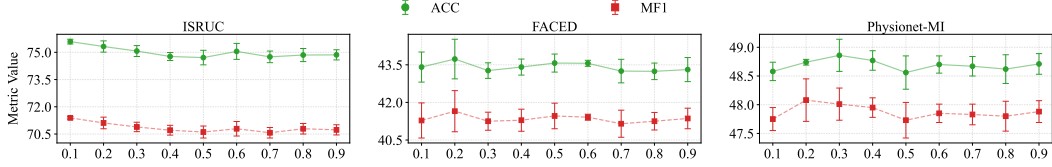

Figure 9: The analysis of hyper-parameter $\alpha$ on different datasets with 30% source domain proportion. The horizontal axis represents the values of $\alpha$, ranging from 0.1 to 0.9 with a step size of 0.1.

## H   Visualization of the Synaptic Network Expansion Process

In this section, we visualize the dynamic synaptic network expansion processes across the ISRUC, FACED, and Physionet-MI datasets under varying source domain proportions, as illustrated in Fig. 10, 11 and Fig. 12. Observations from the visualizations reveal that certain critical nodes during the early stages of synaptic network expansion (i.e., larger node sizes) may gradually lose significance

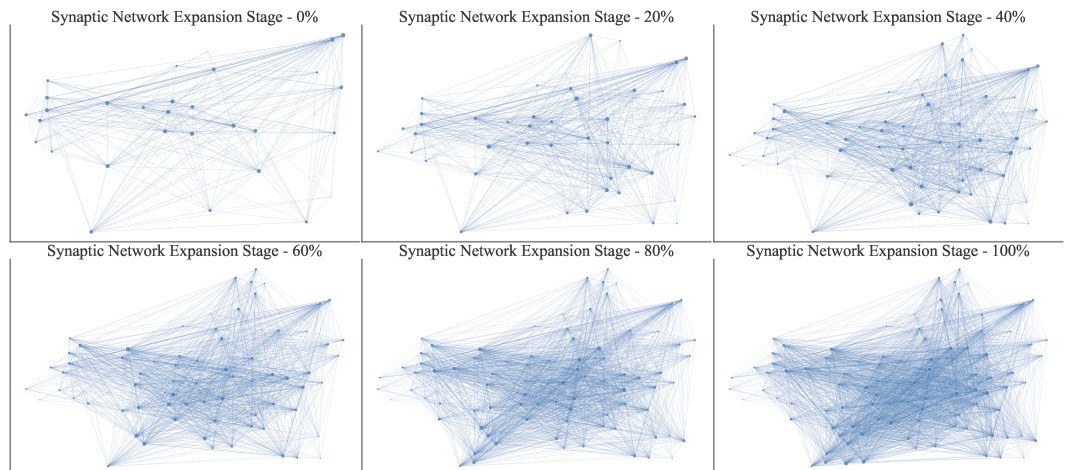

Figure 10: Overview of the synaptic network dynamic expansion process on the ISRUC dataset.

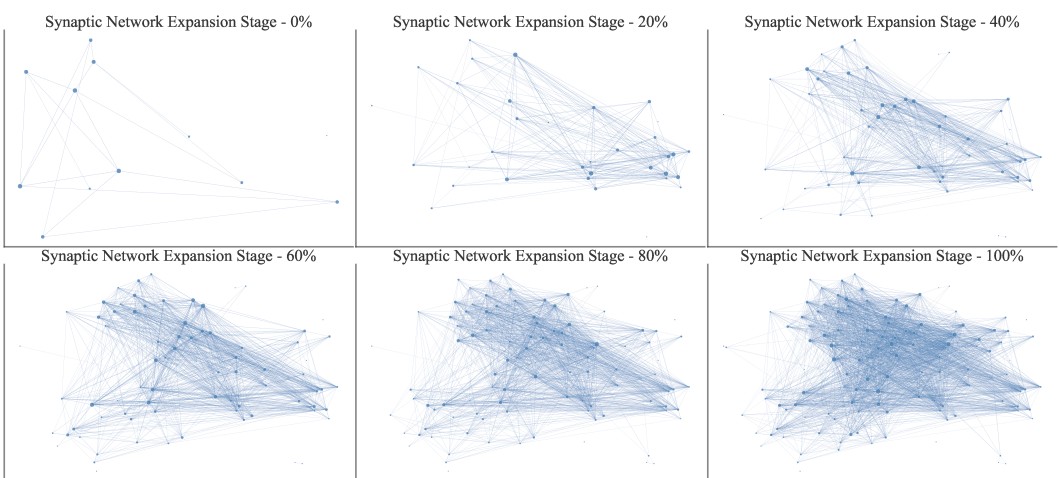

Figure 11: Overview of the synaptic network dynamic expansion process on the FACED dataset.

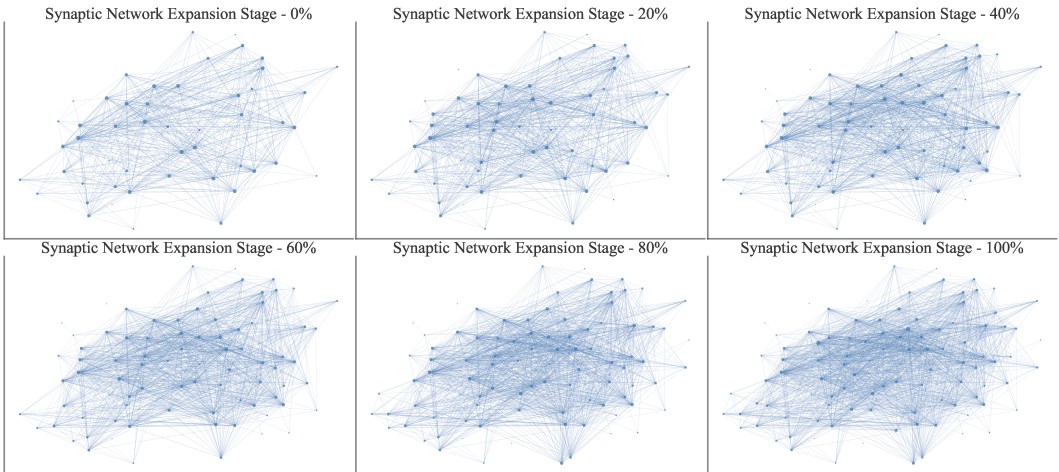

Figure 12: Overview of the synaptic network dynamic expansion process on the Physionet-MI dataset.

as the network expands, triggering synaptic consolidation and renormalization. Conversely, nodes

initially less prominent (i.e., smaller node sizes) may gain increasing importance through progressive integration and connections with newly added nodes. This finding demonstrates that the proposed synaptic network is less susceptible to source domain node initialization, while exhibiting both stability and robustness.

# I Ablation Study

To investigate the effectiveness of synaptic consolidation and synaptic renormalization in SPICED, we conducted an ablation study. The ablated methods are as follows: w/o SR: only synaptic consolidation is adopted; w/o SC: only synaptic renormalization is adopted; SPICED: the framework with all components. Experimental results demonstrate that SPICED's performance degrades in the absence of either synaptic consolidation or synaptic renormalization. This further underscores the critical importance of synaptic homeostasis, arising from the interplay between synaptic consolidation and renormalization, to the SPICED framework—highlighting the indispensability of both mechanisms.

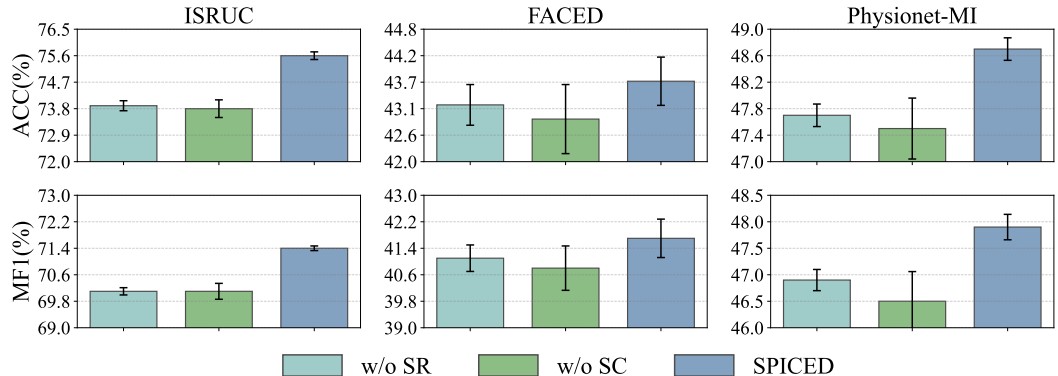

Figure 13: Overview performance comparison with ablated methods at 30% source domain proportions. Each ablated method was evaluated under the same dataset partitioning, with five runs conducted per method by randomly shuffling the input order of the incremental set to enable statistical evaluation.

# J Computational Cost and Resource Requirement

As shown in the Tab. 7, the reported average time per individual includes the full adaptation process: synaptic node incorporation, critical memory reactivation, model training (including synaptic consolidation and weight renormalization) and evaluation. The reported storage refers to the disk storage usage per individual. These results demonstrate that SPICED achieves efficient adaptation with manageable computational overhead. And our model is trained on a single machine equipped with an Intel Core i9 10900K CPU and eight NVIDIA RTX 3080 GPUs.

Table 7: The computational and storage cost per individual.

|                       | ISRUC     | FACED     | Physionet-MI |
| --------------------- | --------- | --------- | ------------ |
| Average Cost (minutes) | 4.42±0.55 | 4.16±0.64 | 4.23±0.63    |
| Storage (M)           | 47.1      | 53.7      | 43.7         |

# K Robustness Study

In this section, we evaluate the robustness of SPICED by adding Gaussian noise scaled to 1%, 5%, and 10% of the original signal's standard deviation during training phase (i.e., progressively noisier conditions). As shown in the Tab. 8, SPICED demonstrates its robustness to such perturbations. Specifically, on ISRUC, SPICED maintains an ACC of 73.8% and MF1 of 70.1% under 10% noise

condition, exhibiting only marginal degradation compared to the clean setting. On FACED, the performance of SPICED remains nearly invariant across all noise levels. And on Physionet-MI, SPICED shows virtually no performance drop across all noise levels, with fluctuations well within the standard error. These results indicate that SPICED is inherently robust to noisy EEG inputs and can effectively tolerate moderate input perturbations without significant performance loss—a desirable property for real-world deployment in low-resource or ambulatory monitoring scenarios.

Table 8: The robustness study of SPICED on three downstream EEG tasks under 30% source domain proportion.

| | ISRUC | | | | FACED | | | | Physionet-MI | | | |
| | ACC | | MF1 | | ACC | | MF1 | | ACC | | MF1 | |
| | $\mathcal{M}_0$ | $\mathcal{M}_i$ | $\mathcal{M}_0$ | $\mathcal{M}_i$ | $\mathcal{M}_0$ | $\mathcal{M}_i$ | $\mathcal{M}_0$ | $\mathcal{M}_i$ | $\mathcal{M}_0$ | $\mathcal{M}_i$ | $\mathcal{M}_0$ | $\mathcal{M}_i$ |
|---|---|---|---|---|---|---|---|---|---|---|---|---|
| 1% Noise | 66.8 | 74.0±0.24 | 60.5 | 70.2±0.21 | 31.7 | 43.1±0.41 | 27.2 | 40.8±0.40 | 42.2 | 48.7±0.14 | 37.9 | 47.8±0.12 |
| 5% Noise | 66.8 | 74.2±0.08 | 60.5 | 70.4±0.21 | 31.7 | 43.2±0.29 | 27.2 | 40.9±0.36 | 42.2 | 48.6±0.14 | 37.9 | 47.8±0.14 |
| 10% Noise | 66.8 | 73.8±0.26 | 60.5 | 70.1±0.24 | 31.7 | 43.0±0.21 | 27.2 | 40.8±0.24 | 42.2 | 48.7±0.05 | 37.9 | 47.8±0.08 |
| Clean EEG | 66.8 | 75.6±0.13 | 60.5 | 71.4±0.07 | 31.7 | 43.7±0.51 | 27.2 | 41.7±0.58 | 42.2 | 48.7±0.17 | 37.9 | 47.9±0.24 |

## L   Discussion

### L.1   Implications

Our work bridges neurobiological principles with artificial continual learning through three key advances: (1) Mechanistic translation—formalizing synaptic homeostasis into a neuromorphic framework that balance stability-plasticity dilemma via biologically grounded consolidation-renormalization dynamics; (2) BCI innovation—enabling individual-specific synaptic network expansion to address inter-individual variability in EEG decoding, supporting continual individual adaptation; (3) Generalizability proof—validated across three different EEG tasks, we demonstrate that bio-inspired synaptic homeostasis mechanism enhances both plasticity for novel individuals and robustness against error accumulation.

### L.2   Limitations and Future Work

In this work, we explores a synaptic homeostasis-inspired framework for unsupervised continual EEG decoding and validates its efficacy across diverse downstream EEG tasks, demonstrating promising performances. However, the following limitations remain: First, modern neuroscience is still at an early stage of understanding the brain, with many neurobiological mechanisms poorly characterized. Consequently, the proposed SPICED framework only approximates the fundamental principles of synaptic homeostasis and does not model deeper neurobiological mechanisms. Second, we adopted a unified initial feature extraction paradigm across different EEG tasks without accounting for inter-task variability in EEG features. Task-specific feature extraction strategies could potentially enable more accurate quantification of inter-individual similarity. In future work, we aim to further investigate neurobiological mechanisms and explore brain-inspired algorithms in broader domains to bridge the gap between biological neural mechanisms and artificial intelligence systems.

