# OpenReview forum: "SPICED: A Synaptic Homeostasis-Inspired Framework for Unsupervised Continual EEG Decoding"
_NeurIPS.cc/2025/Conference — NeurIPS 2025 poster_

### Official Review · Reviewer_HgnD · 2025-07-01

**Clarity:** 1
**Significance:** 3
**Originality:** 2
**Rating:** 4
**Confidence:** 3

**Summary:**

This paper presents a Synaptic Homeostasis-Inspired framework for Unsupervised Continual EEG Decoding (SPICED), which is a biologically inspired continual learning method.   This methodology involves three aspects: i) synaptic consolidation to reinforce critical memory traces, ii) critical memory reactivation to select and replay important traces, and iii) synaptic renormalization to allow learning from new sequences and reduce the effects of detrimental memory traces.

**Questions:**

Questions relate to the strengths and weaknesses above.  In particular:

1. How does this work relate to existing methodologies, especially growing neural gas and self organizing maps?

2. Can you clarify how this relates to standard continual learning scenarios (class incremental, task incremental, etc)?

3. How was the individual data partitioned? Was data for individuals entirely left out for incremental learning?

**Ethical Concerns:**

["NO or VERY MINOR ethics concerns only"]

**Final Justification:**

I would like to thank the authors for their detailed response. Given the clarification on their data preprocessing (training/validation data split) I've updated my rating of the paper.  The issue of similarity with growing neural gas still seems somewhat unresolved as while their method has a biological motivation it does seem the actual algorithm/methodology is quite similar (e.g., cosine similarity can be used as a distance metric for growing neural gas).

**Limitations:**

Yes.

**Paper Formatting Concerns:**

The paper could use an edit for grammar, otherwise it is for the most part well written without any paper formatting problems.

**Quality:**

2

**Strengths And Weaknesses:**

+ The results for this method show better performance to the other methods compared to.

+ Datasets used in the work seem well selected.

- Tests of statistical significance for this method versus the other methods would add strength to the results.

- The methodology seems extremely similar to growing neural gas (a variant of self organizing maps), especially in terms of how nodes are connected and and the proposed “critical connection consolidation” is performed. The paper should be related to work in self organizing maps as much of this appears quite similar.

- The paper claims that this is unsupervised continual learning, however they state that models (for new neurons) are pre-trained on data from new individuals - this is not unsupervised (as it does not appear that new individuals are not detected automatically, rather this information is provided to the methodology), nor does it match standard continual learning scenarios as new individuals are pre-trained on existing data.  It would be good to relate this work to existing continual learning scenarios (task incremental, class incremental, etc)

- It seems that the dataset partitioning is in a sense cherry-picking the problem.  From the paper, each individual’s data is partitioned into a pre-training set and then an incremental set (to use after pretraining), this means the framework has already seen data from all individuals (which also doesn’t line up with continual learning).  Entire individuals should be held out for incremental learning so the performance of the framework could be shown in an actual continual learning scenario.  If individuals were being held out for the incremental version this should be more clearly stated.

---

> ### Author Rebuttal · Authors · 2025-07-31
>
> **We would like to express our sincere gratitude to you for taking the time to review our submission.** In this rebuttal, we will address each of the key issues and points you have raised.
>
> ## Main Comments
>
> > #### **Q1: Tests of statistical significance for this method versus the other methods would add strength to the results.**
>
> **A1: We thank the reviewer for the valuable suggestion.** We have added statistical significance tests comparing SPICED with baseline methods, as summarized in the table below. We will include these results in Section 4.2.2 of the revised manuscript to ensure full transparency and reproducibility.
>
> | 10% | ISRUC (ACC/MF1) | FACED (ACC/MF1) | Physionet-MI (ACC/MF1) |
> | --- | --- | --- | --- |
> | **SPICED vs MMD** | p=0.003** / p=0.002** | p<0.001*** / p<0.001*** | p<0.001*** / p<0.001*** |
> | **SPICED vs EWC** | p=0.048* / p=0.042* | p<0.001*** / p<0.001*** | p=0.38 / p=0.25 |
> | **SPICED vs UCL_GV** | p=0.001** / p<0.001*** | p=0.032* / p=0.026* | p<0.001*** / p<0.001*** |
> | **SPICED vs ReSNT** | p=0.002** / p=0.003** | p<0.001*** / p<0.001*** | p=0.001** / p<0.001*** |
> | **SPICED vs BrainUICL** | p<0.001*** / p<0.001*** | p=0.006** / p=0.003** | p=0.31 / p=0.12 |
>
> | 30% | ISRUC (ACC/MF1) | FACED (ACC/MF1) | Physionet-MI (ACC/MF1) |
> | --- | --- | --- | --- |
> | **SPICED vs MMD** | p<0.001*** / p<0.001*** | p<0.001*** / p<0.001*** | p<0.001*** / p<0.001*** |
> | **SPICED vs EWC** | p<0.001*** / p<0.001*** | p<0.001*** / p<0.001*** | p=0.012* / p=0.008** |
> | **SPICED vs UCL_GV** | p<0.001*** / p<0.001*** | p=0.028* / p=0.019* | p<0.001*** / p<0.001*** |
> | **SPICED vs ReSNT** | p<0.001*** / p<0.001*** | p=0.003** / p=0.002** | p<0.001*** / p<0.001*** |
> | **SPICED vs BrainUICL** | p=0.002** / p=0.003** | p<0.001*** / p<0.001*** | p=0.043* / p=0.038* |
>
> | 50% | ISRUC (ACC/MF1) | FACED (ACC/MF1) | Physionet-MI (ACC/MF1) |
> | --- | --- | --- | --- |
> | **SPICED vs MMD** | p<0.001*** / p<0.001*** | p<0.001*** / p<0.001*** | p<0.001*** / p<0.001*** |
> | **SPICED vs EWC** | p=0.002** / p=0.003** | p=0.682 / p=0.735 | p=0.004** / p=0.002** |
> | **SPICED vs UCL_GV** | p=0.037* / p=0.021* | p<0.001*** / p<0.001*** | p<0.001*** / p<0.001*** |
> | **SPICED vs ReSNT** | p=0.008** / p=0.005** | p<0.001*** / p<0.001*** | p<0.001*** / p<0.001*** |
> | **SPICED vs BrainUICL** | p=0.019* / p=0.007** | p<0.001*** / p<0.001*** | p=0.412 / p=0.683 |
>
> ---
>
> > #### **Q2: How does this work relate to existing methodologies, especially growing neural gas and self organizing maps?**
>
> **A2: We thank the reviewer for raising this profound and crucial question regarding the relationship between SPICED and classical models such as Growing Neural Gas (GNG) and Self-Organizing Maps (SOM).** We have carefully read the relevant literature on GNG and SOM and summarize our responses below:
>
> While SPICED shares similarities with these methods in that all involve dynamic, topology-preserving networks, our work is fundamentally distinct in **biological motivation, functional mechanism, and application objective**.
>
> 1. **Biological Motivation**
>
>   - While GNG and SOM are neurally inspired algorithms for topological mapping and vector quantization, they do not explicitly model specific neurobiological learning mechanisms.
>
>   - In contrast, SPICED is **directly grounded in synaptic homeostasis**—a well-established biological principle involving the dynamic interplay between synaptic consolidation and renormalization during learning and sleep.
>
>     Our framework explicitly emulates three key aspects of brain mechanism: **synaptic consolidation**, **synaptic renormalization**, and the **brain’s functional specificity**, integrating them into a principled and biologically plausible continual learning framework. By doing so, SPICED achieves:
>
>     - Biologically plausible memory stabilization
>
>     - Adaptive forgetting of non-essential information
>
>     - Maintenance of network plasticity for continual learning
>
>
>   This **mechanistic translation of neurobiology into a computational framework** is a **core contribution** of our work. It represents a key advance over traditional self-organizing approaches, enabling more human-like stability-plasticity balance in artificial continual learning systems.
>
> 2. **Functional Mechanism**
>
>   |     | GNG | SPICED |
>   | --- | --- | --- |
>   | **Biological Mechanism** | Neurally inspired, general topology learning | Explicitly models **synaptic homeostasis** |
>   | **Network Growth** | Error-driven expansion for **density approximation** | **Individual-specific node** integration for continual individual adaptation |
>   | **Connection Mechanism** | Based on proximity (e.g., nearest neighbor) | Based on **initial feature similarity** (weighted cosine similarity) |
>   | **Memory Replay** | None | **Importance-weighted replay** of task-discriminative memories |
>   | **Model Fusion** | N/A | **Weighted aggregation of historical models**, not sequential updates |
>   | **Stability Mechanism** | Edge aging / neighborhood decay | **Periodic synaptic renormalization** with time-dependent decay |
>
>
>        Unlike traditional continual learning or GNG-style adaptation, SPICED **does not rely on the previous model state ​**. Instead, it **integrates information across the entire synaptic network**, making it robust to outlier individuals and catastrophic forgetting.
>
> 3. **Application Objectives**
>
>   - **Primary Goal:** GNG and SOM are general-purpose unsupervised learning tools for clustering and visualization. SPICED is designed for a specific and challenging BCI problem: unsupervised continual EEG decoding under inter-individual variability.
>   - **Key Challenge:** While GNG/SOM primarily address static data topology organization, SPICED is explicitly motivated by the dynamic stability-plasticity balance observed in biological neural systems, and thus targets two fundamental challenges in real-world continual EEG decoding: **(1)** robust adaptation to the sequential arrival of new individuals under significant domain shifts, and **(2)** mitigation of catastrophic forgetting during long-term learning. This design is directly inspired by synaptic homeostasis—the brain’s mechanism for integrating novel experiences while preserving consolidated memories.
>
> **Thanks once again for your insightful comment.** We have added a discussion comparing SPICED with GNG and SOM in the revised manuscript, clarifying their conceptual relationships and differences. We hope this addresses your concern.
>
> ---
>
> > #### **Q3: From the paper, each individual’s data is partitioned into a pre-training set and then an incremental set (to use after pretraining), this means the framework has already seen data from all individuals. How was the individual data partitioned? Was data for individuals entirely left out for incremental learning?**
>
> **A3: We thank the reviewer for raising this question.** There might be a misunderstanding regarding our experimental setup. Our protocol is **subject-independent**, meaning that the individuals involved in the incremental phase never appear during pre-training. The incremental learning process is conducted in a completely **unsupervised manner**. For instance, in Table 2, the 10% split indicates that 10% of the subjects in the entire dataset are used as the source domain for pre-training the model, while the remaining 90% of subjects constitute the incremental stream for long-term continual learning. To prevent confusion, we have revised the description of the dataset split in Section 4.1 as follows.
>
> > "Each dataset is partitioned in a **subject-independent manner**: a subset of subjects forms the pretraining set (i.e., source domain), used solely for pre-training the source model, while the remaining subjects constitute the incremental set (i.e., target domain), which is used to evaluate the performance of the SPICED framework in unsupervised individual continual learning (i.e., continual EEG decoding) scenario"
>
> **We appreciate the reviewer pointing this out, and we hope this clarification adequately addresses your concern.**
>
> ---
>
> > #### **Q4: Can you clarify how this relates to standard continual learning scenarios (class incremental, task incremental, etc)?**
>
> **A4: Thank you for this insightful question.** Our experimental paradigm is based on **Unsupervised Individual Continual Learning (UICL)**, which falls under the broader category of **domain-incremental learning** in continual learning. In this setting, the task (e.g., sleep staging, emotion recognition) remains fixed, while new individuals—each representing a distinct data domain—arrive sequentially. The core challenge arises from **significant inter-individual domain shifts**, rather than changes in label space (as in class-incremental) or task identity (as in task-incremental) scenarios.
>
> Notably, unlike traditional continual learning paradigms that rely on the incremental evolution of a single model (i.e., $M_i$ derived from $M_{i-1}$ ), SPICED adopts a fundamentally different strategy. When adapting to a new individual, SPICED does not depend on the model state from the previous timestep. Instead, it **integrates information across the entire synaptic network** by:
>
> - Selectively reactivating **task-discriminative memory traces** (i.e., critical memory replay), and
> - Performing **weighted model fusion** over top-K relevant historical models (i.e., critical model fusion).
>
> This global integration mechanism enables SPICED to prioritize robust, task-relevant knowledge while suppressing interference from outlier or noisy domains. As a result, it achieves **robust adaptation** even in the presence of large domain shifts, and effectively **mitigates catastrophic forgetting** over long-term learning sequences.
>
> **We sincerely thank the reviewer for their careful reading and thoughtful feedback.** We hope our clarification contributes to a clearer and more complete understanding of our work.

---

### Official Review · Reviewer_zYsa · 2025-07-02

**Clarity:** 3
**Significance:** 2
**Originality:** 2
**Rating:** 4
**Confidence:** 4

**Summary:**

The authors propose a biologically inspired method for continually learning framework called SPICED from EEG signals that balances learning new information, preserving important past memories. This framework has three key processes: critical memory reactivation that replays relevant past knowledge, synaptic consolidation that strengthens important connections, and finally synaptic renormalization that weakens and reduces the plasticity of less useful connections, enabling adaptability and memory retention. It represents each individual as a node in a dynamic synaptic network, connecting new individuals to similar past ones based on EEG feature similarity by fusing information from multiple relevant models and trains on pseudo-labelled data, The framework is validated on three diverse EEG tasks: sleep staging, emotion recognition, and motor imagery.

**Questions:**

•Can you report forward transfer (FWT) and backward transfer (BWT) which are standard metrics in continual learning for quantifying positive transfer and forgetting? Including these would strengthen your empirical claims and enable fairer comparison with existing literature. Additionally, can you evaluate SPICED against more recent continual learning methods?

•The paper uses mechanisms like exponential decay and weighted averaging to simulate synaptic forgetting and fusion. How closely do these mechanisms align with actual neural processes? Can you clarify which aspects of synaptic homeostasis are modelled vs abstracted, and cite supporting neuroscience literature.

•Can the authors provide explanations or visualizations of the synaptic trace evolution, model fusion weights, or importance scores? Additionally how does SPICED model perform when the source individual pool is small, noisy, or low quality?

•Can SPICED generalize to non-EEG domains such as sensor data, speech, or time-series forecasting? Please discuss how the core mechanisms such as synaptic network, consolidation, renormalization might transfer to other modalities, and what adaptations might be required. Addressing this would broaden the significance and applicability of your approach.

**Ethical Concerns:**

["NO or VERY MINOR ethics concerns only"]

**Final Justification:**

After closely examining the rebuttal and other reviewers’ comments, I conclude that most of my concerns have been addressed; accordingly, I will revise my score to Borderline Accept.

**Limitations:**

Yes

**Quality:**

2

**Strengths And Weaknesses:**

Strengths:

•SPICED framework is based on synaptic homeostasis demonstrating both synaptic consolidation (strengthening) and renormalization (forgetting), supporting selective replay, model fusion, and adaptive forgetting, without modifying the core model architecture.

•The framework introduces a synaptic network, where each node represents an individual, edges reflecting similarity and memory strength, allowing to selectively reactivate and replay task relevant past individuals based on importance scores combining similarity and synaptic strength.

•It initializes new models by fusing multiple past models using importance-weighted aggregation and simulates forgetting using time dependent synaptic decay, stimulating biologically plausible memory dynamics.

•The framework simulates biologically plausible forgetting through time-dependent synaptic decay and provides interpretable outputs like synaptic strength trajectories, network visualizations that  help understand what the model remembers and forgets during continual learning.

•The framework has been validated on three EEG datasets: ISRUC (sleep), FACED (emotion), and PhysioNet-MI (motor imagery) with ablation studies, hyperparameter analysis, and visualizations of memory dynamics.

Weaknesses:

•Although SPICED is presented as an unsupervised CL method, it relies on labelled source-domain individuals for initialization and pretraining. This makes it semi-supervised in practice, and the unsupervised claim is misleading without further clarification.

•The paper does not report forward transfer (FWT) or backward transfer (BWT) metrics, which are widely used in continual learning to quantify learning and forgetting. Their inclusion would offer deeper insight into SPICED’s stability plasticity behavior.

•Although biologically inspired, the implementation simplifies complex neural phenomena as an example, by using exponential decay for forgetting, model fusion as weighted averaging. The biological claims are not validated against neuroscience literature.

•The effectiveness of SPICED depends on the quality and diversity of the labelled source individuals and the performance under low-resource or poor initialization scenarios is not deeply analyzed.

•Most baselines are either EEG-specific or older CL methods like EWC and MMD. The paper does not compare against recent strong replay-based or dynamic CL methods, including more recent continual learning techniques (e.g., replay-based or dynamic architectures) would strengthen claims.

---

> ### Author Rebuttal · Authors · 2025-07-31
>
> **We'd like to express our sincere gratitude for your careful readings and helpful comments.** We provide our feedbacks as follows.
>
> ## Main Comments
>
> > #### **Q1: SPICED relies on labelled source-domain individuals for initialization and pretraining. The unsupervised claim is misleading without further clarification.**
>
> **A1: Many thanks for pointing out this.** We agree and we have refined the part of Introduction to better contextualize the unsupervised claim and avoid potential misinterpretation. Specifically, as stated in Section 1 (first paragraph):
>
> > "...especially in non-stationary continual EEG decoding scenarios **where the pretrained source model is required to adapt to unseen individuals continuously.**"
>
> ---
>
> > #### **Q2: Can you report forward transfer (FWT) and backward transfer (BWT) which are standard metrics in continual learning for quantifying positive transfer and forgetting?**
>
> **A2: We thank the reviewer for the valuable suggestion.** However, FWT and BWT are ill-suited for the individual continual learning setting, which fundamentally differs from standard CL process(e.g., class-incremental, task-incremental). The reasons are as follows:
>
> 1. **Different Core Objectives:** SPICED addresses individual continual learning, not standard CL scenarios (e.g., class- or task-incremental learning) where models must retain performance on discrete, fixed tasks. In real-world deployment, **the goal is not to memorize every past individual, but to robustly adapt to each new subject** while mitigating error accumulation—precisely the stability-plasticity balance achieved through synaptic homeostasis. Our focus is on continual personalization, not task-level transfer. The core challenge is preventing catastrophic forgetting of learning capacity, not preserving fixed task outputs. SPICED meets this via dynamic synaptic consolidation and renormalization, enabling sustained adaptation across individuals—without requiring perfect recall of prior ones.
>
> 2. **Different Model Architecture:** In traditional CL, a single backbone model undergoes incremental adaptation across tasks, enabling BWT/FWT to measure performance changes on a fixed model. In contrast, SPICED employs a dynamically expanding synaptic network, where each individual has a dedicated node (storing model, connections). The current model is derived by fusing **global historical information** via importance-weighted aggregation—not by sequential fine-tuning of a single backbone. Consequently, evaluating this fused model on past or future individuals is not meaningful. Our evaluation—measuring adaptation gain and long-term stability—directly reflects the framework’s goal: continual personalization under distribution shift, not task-level transfer or fixed-model retention.
>
> **Thanks once again for your constructive suggestions. We hope our clarification adequately addresses your concerns.**
>
> ---
>
> > #### **Q3: Can you clarify which aspects of synaptic homeostasis are modelled vs abstracted, and cite supporting neuroscience literature.**
>
> **A3: Thank you for raising this profound and crucial question.** We fully agree that biological plausibility is a central concern for brain-inspired computing frameworks. In SPICED, the employed exponential decay and weighted averaging mechanisms are not arbitrary designs, but rather principled computational abstractions of key neural mechanisms underlying synaptic homeostasis. These approaches are grounded in well-established biological foundations, as detailed below:
>
> 1. **Exponential Decay based Synaptic Renormalization:** We emphasize that the exponential decay mechanism (Eq. 6) is directly inspired by the **Ebbinghaus forgetting curve [1]**. Neuroscientific studies have established that during slow-wave sleep, the brain undergoes a global, proportional weakening of synaptic strength—a process known as **synaptic downscaling [2]**, which serves the critical function of preventing synaptic saturation and restoring learning capacity. Our exponential decay formulation models this **global downscaling effect**, with its time-dependent decay rate aligned with the Ebbinghaus curve, a well-validated model of natural memory decay [1]. Importantly, this is not merely a heuristic choice but rather a **principled mathematical characterization** of the passive, time-dependent weakening of synaptic strength observed in biological systems.
>
> 2. **Critical Memory Reactivation based Weighted Fusion:** The weighted averaging in model fusion (Eq. 3) is not ad hoc, but a principled abstraction of the **brain’s functional specificity [3]**, where task-relevant neural ensembles are selectively recruited rather than uniformly activating all memories. Our importance coefficient (Eq. 2) combines individual similarity (initial feature similarity) and historical activation strength (average connection strength). The former models representation-based memory retrieval, while the latter reflects the **cumulative synaptic plasticity state (e.g., LTP/LTD effects) [4]**. This “relevance + stability”-based weighting constitutes a biologically grounded mechanism for selective information integration, consistent with neurocognitive principles.
>
> We do not simulate molecular-level ion channel dynamics or precise spike timing, as our goal is to build a scalable, functionally equivalent computational framework—not a biological simulator. The LTP and LTD mechanisms constitute the core computational primitives of synaptic plasticity. **We abstract these mechanisms into general principles for continual learning, preserving biological plausibility while effectively addressing practical challenges in continual EEG decoding.**
>
> [1] Hermann Ebbinghaus. Über das gedächtnis: untersuchungen zur experimentellen psychologie. Duncker & Humblot, 1885.
>
> [2] Giulio Tononi and Chiara Cirelli. Sleep and synaptic homeostasis: a hypothesis. Brain research bulletin, 62(2):143–150, 2003.
>
> [3] Karl Friston. A theory of cortical responses. Philosophical transactions of the Royal Society B:Biological sciences, 360(1456):815–836, 2005.
>
> [4] Robert C Malenka, Nicoll, and Roger A. Long-term potentiation–a decade of progress? Science, 285(5435):1870–1874, 1999.
>
> ---
>
> > #### **Q4: Can the authors provide explanations or visualizations of the synaptic trace evolution, model fusion weights, or importance scores?**
>
> **A4: We thank the reviewer for this valuable question.** Visualizations and analyses of synaptic trace evolution and average synaptic connection strength are already provided in **Appendix H and I (page18-19)**, respectively. Due to submission constraints in the rebuttal phase, we are unable to include figures here. Regarding model fusion weights and importance scores, we will include corresponding visualizations and analysis in the appendix of the revised manuscript to further clarify their evolution.
>
> ---
>
> > #### **Q5: The effectiveness of SPICED depends on the quality and diversity of the labelled source individuals and the performance under low-resource or poor initialization scenarios is not deeply analyzed.**
>
> **A5: Thank for your meaningful comment.** We have indeed evaluated SPICED under low-resource and poor initialization settings, as detailed in **Table 2 (page 8)**. Specifically, we test source domain proportions of 10%, where labeled source individuals are extremely limited (with incremental individuals outnumbering source ones by 9×). This setting leads to **severely degraded initial model performance** (e.g., $M_0$ ACC = 56.8% on ISRUC at 10%), reflecting a challenging poor-initialization regime. However, under such conditions, SPICED achieves **consistent performance gains** after adaptation (e.g., +5.4 ACC and +7.0 MF1 on ISRUC), while most baseline methods exhibit significant performance degradation when pretrained with only 10% source data, as shown in Figure 4 (page8).
>
> ---
>
> > #### **Q6: Can SPICED generalize to non-EEG domains such as sensor data, speech, or time-series forecasting?**
>
> **A6: We thank the reviewer for this insightful question regarding the generalizability of SPICED.** While our work focuses on EEG decoding, the core mechanisms of SPICED—synaptic network construction, critical memory reactivation, consolidation, and renormalization—are **modality-agnostic** principles inspired by universal neural computation mechanisms, and thus hold strong potential for transfer to other sequential, individualized, or domain-shift-prone settings. For instance: In **wearable sensor data** (e.g., activity recognition), individuals exhibit high variability in motion patterns. SPICED could maintain a synaptic network of past users, selectively reactivating similar ones to bootstrap adaptation for a new user. Similarly, SPICED can be generalized to other specific scenarios where a sequential, individualized feature space exists. We will incorporate this discussion into the revised manuscript.
>
> ---
>
> > #### **Q7: The paper does not compare against recent replay-based or dynamic CL methods. Can you evaluate SPICED against more recent continual learning methods?**
>
> **A7: Thanks for your helpful suggestion.** We have incorporated **CoUDA**, a recent method for continual domain adaptation, as an additional baseline.; due to space constraints in the rebuttal, we refer the reviewer to **Response A3 for Reviewer# yKhn**, which includes a detailed comparison table and analysis. Furthermore, we clarify that the compared methods ReSNT and BrainUICL are not generic CL or EEG-specific methods. **They are dynamic replay-based approaches designed for continual EEG decoding**, and published in 2023 and 2025, respectively. Our comparison is thus grounded in the most relevant and up-to-date literature for this specific problem setting.
>
> **We sincerely thank you again for your constructive comments.** We hope that this rebuttal has adequately addressed your concerns, and we would be happy to provide further clarification if you have any additional questions.

---

> > ### Author Response · Authors · 2025-08-05
> > **Rebuttal has been submitted and we are eager to hear your further constructive feedback**
> >
> > **Dear Reviewer zYsa**
> >
> > **We hope this message finds you well. We sincerely appreciate your thoughtful and insightful feedback.** In response, we have made the following clarifications to address your concerns:
> >
> > 1. **Unsupervised claim:** We further clarified the scope of "unsupervised" in the Introduction to avoid misinterpretation.
> >
> > 2. **Continual learning metrics:** We have explained why FWT/BWT are not applicable in individual continual learning settings.
> >
> > 3. **Biological plausibility:** We further clarified the neuroscience foundations of synaptic homeostasis in SPICED with supporting references.
> >
> > 4. **Low-resource performance:** We have highlighted SPICED’s robustness under poor initialization (e.g., 10% source data) with quantitative results.
> >
> > 5. **Generalizability of SPICED**: We have discussed SPICED’s potential extension to non-EEG domains in the revised manuscript.
> >
> > 6. **Baseline comparisons**: We have added CoUDA as a recent baseline and justified the relevance of existing compared methods.
> >
> > **We hope these responses clearly address your valuable comments.** Given the review timeline, we kindly encourage you to consider our rebuttal when finalizing your evaluation. **Should you have any further comments or suggestions, please do not hesitate to reach out. We remain available throughout the revision process.**
> >
> > Thank you again for your constructive feedback.
> >
> > Best regards,
> >
> > The authors

---

> > > ### Author Response · Authors · 2025-08-08
> > >
> > > **Dear Reviewer zYsa:**
> > >
> > > We hope this message finds you well. **As the discussion period is nearing its end with less than one day remaining, we wanted to ensure we have addressed all your concerns satisfactorily**. If there are any additional points or feedback you'd like us to consider, please let us know. Your insights are invaluable to us, and we're eager to address any remaining issues to improve our work.
> > >
> > > **Thank you again for your time and effort in reviewing our paper and providing constructive feedbacks.**
> > >
> > > Best regards,
> > >
> > > The authors

---

### Official Review · Reviewer_yKhn · 2025-07-03

**Clarity:** 3
**Significance:** 3
**Originality:** 3
**Rating:** 5
**Confidence:** 3

**Summary:**

This paper presents a biologically inspired continual learning framework for EEG decoding, motivated by synaptic homeostasis principles. The model incrementally adapts by: (1) initializing a base network, (2) incorporating new neurons based on pairwise similarity with existing representations, (3) consolidating new and old knowledge through model fusion, and (4) applying renormalization for adaptability. These steps are clearly explained and visually illustrated. The method is evaluated on three EEG datasets and compared against several recent baselines (e.g., MMD, EWC, UCL-GV, ReSNT, BrainUICL) using metrics like accuracy and macro-F1. The results demonstrate competitive performance, and the authors include ablation studies to support design choices.

**Questions:**

1. Could you briefly describe the baseline methods (e.g., BrainUICL, UCL-GV, ReSNT) in the related work section to better frame your contribution? A comparison of what their identified gaps are and how your method takes it forward.
2. Are all hyperparameters (including learning rates, weight terms, and consolidation thresholds) tuned individually for each dataset? If so, how feasible is this setup for real-world deployment?
3. Could you provide a main table with the exact numerical values for accuracy and macro-F1, including standard deviations or confidence intervals, instead of only bar plots?

**Ethical Concerns:**

["NO or VERY MINOR ethics concerns only"]

**Final Justification:**

My initial borderline accept rating stemmed from concerns about missing baseline descriptions and gap analysis, the tuning of hyperparameters per dataset and its real-world feasibility, and the lack of a main table with exact numerical results and variability measures. In the rebuttal, the authors provided detailed clarifications, including a direct comparison against the most relevant recent baseline (CoUDA, 2025), where their method outperforms on 2 of the 3 datasets. They also explained that their approach does not rely on previous inputs like other methods, instead leveraging global statistical patterns, and showed that only one hyperparameter was tuned per dataset while others remained fixed - strengthening claims of robustness. These responses address my main doubts and increase my confidence in the empirical validity of the work. Given the novelty, practical applicability, and clarified experimental rigor, I am raising my score to an accept.

**Limitations:**

Yes.

**Paper Formatting Concerns:**

No formatting concerns.

**Quality:**

3

**Strengths And Weaknesses:**

Strengths:

1. The paper is very well-written and organized, with each methodological step presented clearly in logical subsections.
2. The main algorithm is biologically intuitively explained and supported by a clear diagram that improves understanding.
3. The evaluation is comprehensive: the proposed approach is compared against several relevant baselines and tested across three EEG datasets.
4. The proposed method is more real-world oriented and this makes the work practically meaningful and applicable beyond standard benchmarks.

Weaknesses:

1. The related work section does not contextualize the compared baselines (e.g., BrainUICL, ReSNT, UCL-GV); instead, these are only introduced in the experiments. Briefly describing them earlier would better highlight the novelty and motivation of the proposed method. As in what is the gap in the other methods that this work is filling.
2. The performance metrics (accuracy and macro-F1) are not defined anywhere. A brief explanation would be helpful for readers unfamiliar with these metrics.
3. The main comparison table against the recent benchmarks is shown only as bar plots, without numerical values, standard deviations, or statistical significance reporting, which limits interpretability and reproducibility.
4. It seems like a lof of tunable hyperparameters according to the appendix. It's unclear whether all the 10-15 hyperparameters mentioned are tuned exhaustively or selectively.

---

> ### Author Rebuttal · Authors · 2025-07-31
>
> **Many thanks for the your detailed and insightful comments. We are glad to see you approved the contributions of our work.** We provide our feedbacks as follows:
>
> ## Main Comments
>
> > #### **Q1: Could you briefly describe the baseline methods (e.g., BrainUICL, UCL-GV, ReSNT) in the related work section to better frame your contribution? A comparison of what their identified gaps are and how your method takes it forward.**
>
> **A1: Thanks for your valuable comments.** We have revised a portion of the Related Work to emphasize our contributions, highlighting the following points:
>
> 1. **Limitations of Traditional Continual EEG Decoding Methods:** "(Sec.2.3) ... in clinics. This limitation has motivated recent efforts to explore individual-specific continual EEG decoding algorithms. UCL-GV [1] employs a FIFO-based buffer and contrastive alignment strategies to mitigate domain shift. ReSNT [2] introduces a dynamically evolving replay strategy for continuous EEG decoding. BrainUICL [3] proposes an unsupervised individual continual learning framework to address the stability-plasticity dilemma in CL. **However, these methods—grounded in incremental model evolution—exhibit strong dependence on the model state from the previous time step**, limiting their ability to cope with large, sustained domain shifts."
> 2. **Innovations of SPICED:** "(Sec.2.3)... domain shifts. By contrast, **SPICED aggregates global historical information for decision-making** when adapting to new tasks and activates the most relevant memories to assist adaptation. It overcomes the limitations of incremental model evolution, ensuring robustness against domain shifts while maintaining performance."
>
> [1] Abu Md Niamul Taufique, Chowdhury Sadman Jahan, and Andreas Savakis. Unsupervised continual learning for gradually varying domains. In Proceedings of the IEEE/CVF conference on computer vision and pattern recognition, pages 3740–3750, 2022.
>
> [2] Tiehang Duan, Zhenyi Wang, Gianfranco Doretto, Fang Li, Cui Tao, and Donald Adjeroh. Replay with stochastic neural transformation for online continual eeg classification. In 2023 IEEE International Conference on Bioinformatics and Biomedicine (BIBM), pages 1874–1879.IEEE, 2023.
>
> [3] Yangxuan Zhou, Sha Zhao, Jiquan Wang, Haiteng Jiang, Shijian Li, Tao Li, and Gang Pan. Brainuicl: An unsupervised individual continual learning framework for eeg applications. In The Thirteenth International Conference on Learning Representations.
>
> ---
>
> > #### **Q2: The performance metrics (accuracy and macro-F1) are not defined anywhere.**
>
> **A2: We thank the reviewer for the valuable suggestion.** We have added the detailed definitions of accuracy and macro-F1 metrics in Section 4.1 as follows:
>
> $$
> \text{Accuracy} = \frac{1}{N} \sum_{i=1}^{N} \mathbb{1}(y_i = \hat{y}_i)
> $$
>
> where $N$ is the total number of samples, $y_i$ is the true label, $\hat{y}_i$ is the predicted label, and $\mathbb{1}(\cdot)$ is the indicator function.
>
> $$
> \text{Macro-F1} = \frac{1}{C} \sum_{c=1}^{C} \frac{2 \cdot \text{Precision}_c \cdot \text{Recall}_c}{\text{Precision}_c + \text{Recall}_c}
> $$
>
> where $C$ is the number of classes, and $\text{Precision}_c$ and $\text{Recall}_c$ are the precision and recall for class $c$, respectively.
>
> ---
>
> > #### **Q3: Could you provide a main table with the exact numerical values for accuracy and macro-F1, including standard deviations or confidence intervals, instead of only bar plots?**
>
> **A3: Many thanks for your constructive suggestions.** In response, we have replaced Figure 4 with a tabular format to better present the results, as shown below. Due to space constraints in the rebuttal, we have split the table into three parts for clarity.
>
> |     | **ISRUC** |     | **FACED** |     | **Physionet-MI** |     |
> | --- | --- | --- | --- | --- | --- | --- |
> | **10%** | ACC | MF1 | ACC | MF1 | ACC | MF1 |
> | MMD | 57.0±0.47 | 50.0±0.65 | 24.1±0.24 | 20.9±0.34 | 37.3±0.34 | 34.3±0.49 |
> | EWC | 60.6±0.71 | 54.1±0.62 | 22.3±0.36 | 19.0±0.49 | 42.8±1.19 | 40.6±2.19 |
> | UCL_GV | 52.5±1.16 | 43.4±1.07 | 25.0±0.43 | 21.9±0.51 | 33.1±0.35 | 24.8±0.51 |
> | ReSNT | 56.4±0.70 | 49.3±1.12 | 14.1±1.66 | 7.1±2.56 | 35.9±1.49 | 30.4±1.79 |
> | BrainUICL | 56.9±0.46 | 48.4±0.68 | 21.2±2.04 | 16.9±2.55 | 43.1±0.33 | 41.5±0.36 |
> | **CoUDA** | 52.7±0.44 | 44.5±0.48 | 23.9±0.16 | 20.5±0.17 | 41.6±0.25 | 38.7±0.39 |
> | **SPICED** | **62.6±1.35** | **55.4±1.51** | **25.7±0.10** | **22.7±0.17** | **43.4±0.44** | **42.0±0.46** |
>
> |     | **ISRUC** |     | **FACED** |     | **Physionet-MI** |     |
> | --- | --- | --- | --- | --- | --- | --- |
> | **30%** | ACC | MF1 | ACC | MF1 | ACC | MF1 |
> | MMD | 70.2±0.83 | 64.9±0.83 | 39.0±1.48 | 35.8±1.66 | 44.1±0.38 | 41.7±0.57 |
> | EWC | 71.3±0.49 | 67.3±0.40 | 39.5±0.94 | 37.0±1.39 | 47.3±0.58 | 45.2±0.75 |
> | UCL_GV | 72.5±0.32 | 68.1±0.45 | 41.4±1.50 | 38.9±1.59 | 39.7±0.22 | 35.8±0.40 |
> | ReSNT | 71.3±0.89 | 67.7±0.78 | 33.3±8.10 | 29.1±9.74 | 44.1±0.50 | 43.1±0.49 |
> | BrainUICL | 74.8±0.11 | 70.8±0.25 | 33.8±2.90 | 29.7±3.55 | 48.3±0.38 | 46.8±0.38 |
> | **CoUDA** | 73.3±0.23 | 69.2±0.23 | 39.2±1.06 | 36.2±1.05 | 44.2±0.31 | 41.1±0.43 |
> | **SPICED** | **75.6±0.13** | **71.4±0.07** | **43.7±0.51** | **41.7±0.58** | **48.7±0.17** | **47.9±0.24** |
>
> |     | **ISRUC** |     | **FACED** |     | **Physionet-MI** |     |
> | --- | --- | --- | --- | --- | --- | --- |
> | **50%** | ACC | MF1 | ACC | MF1 | ACC | MF1 |
> | MMD | 68.2±1.00 | 62.2±1.37 | 19.1±2.59 | 10.0±3.09 | 47.9±0.54 | 45.9±0.63 |
> | EWC | 72.5±0.39 | 67.2±0.80 | 48.7±0.66 | 46.7±1.63 | 50.2±0.43 | 48.0±0.64 |
> | UCL_GV | 73.9±0.29 | 68.4±0.26 | 22.8±3.64 | 14.4±4.33 | 45.5±0.71 | 42.6±0.58 |
> | ReSNT | 72.1±0.69 | 66.5±0.77 | 23.6±3.29 | 15.5±3.96 | 48.8±0.28 | 45.7±0.48 |
> | BrainUICL | 73.6±0.29 | 67.7±0.29 | 20.2±3.46 | 12.0±3.92 | **52.5±0.36** | **51.3±0.48** |
> | **CoUDA** | 72.3±0.94 | 66.4±0.73 | 22.5±2.61 | 13.7±3.31 | 51.4±0.16 | 49.6±0.16 |
> | **SPICED** | **74.5±0.39** | **69.2±0.37** | **48.9±0.42** | **47.0±0.50** | 52.4±0.11 | 51.2±0.14 |
>
>      Furthermore, we have incorporated **CoUDA[1]**, a recent method for continual domain adaptation, as an additional baseline. As shown in the results, SPICED consistently outperforms existing approaches across various data splits, demonstrating strong stability and robustness. For instance, under low-resource settings, most methods suffer performance degradation compared to their initial performance, whereas SPICED is able to further enhance model adaptability even under poor initialization. On the 50% split of FACED, where significant performance drops are observed for most methods—likely due to continuous and substantial domain shifts—SPICED maintains robust performance. This is attributed to its critical memory reactivation mechanism that **aggregates global historical information**, effectively mitigating the impact of domain shifts and improving model adaptability across diverse individuals.
>
> **We thank the reviewer once again for this valuable suggestion.** We hope that this revision provides a clearer overview and addresses your concern.
>
> [1]Chen B, Zhang X, Shen C, et al. CoUDA: Continual Unsupervised Domain Adaptation for Industrial Fault Diagnosis Under Dynamic Working Conditions[J]. IEEE Transactions on Industrial Informatics, 2025.
>
> ---
>
> > #### **Q4: Are all hyperparameters (including learning rates, weight terms, and consolidation thresholds) tuned individually for each dataset? If so, how feasible is this setup for real-world deployment?**
>
> **A4: We thank the reviewer for the valuable and insightful question.** To clarify, **all the hyperparameters**—including learning rates, weight terms, and other optimization settings—are **kept consistent across all datasets**. The **only exception is the synaptic connection threshold $\xi$** , which is adjusted according to the intrinsic characteristics of each dataset, as detailed in Section 4.2.1 and Appendix F.
>
> The rationale for this adjustment is grounded in the nature of the data:
>
> - For **resting-state datasets** such as ISRUC, significant inter-individual variability in initial individual features necessitates a **lower threshold** $\xi$ to ensure sufficient synaptic connection density, thereby preserving meaningful individualized patterns.
> - In contrast, for **task-based datasets** like FACED and PhysioNet-MI, **participants are exposed to identical stimuli**, leading to more homogeneous initial representations across individuals. In this case, a **higher threshold** $\xi$ is adopted to filter out redundant, overly similar connections while retaining those that are discriminative at the individual level.
>
> The adaptive setting of the connection threshold involves only a lightweight, one-time calibration based on data modality (resting-state vs. task-based), which is justified since high-level experimental conditions—such as task design—are typically known prior to deployment. Importantly, all the other hyperparameters are kept fixed across datasets and experiments. Given this minimal adaptation and consistent configuration, our approach **remains practical and highly feasible for real-world deployment scenarios.**
>
> **We hope this clarification addresses the reviewer’s concern. We thank the reviewer once again for your thoughtful and valuable feedback.**

---

> > ### Author Response · Authors · 2025-08-05
> > **Rebuttal has been submitted and we are eager to hear your further constructive feedback**
> >
> > **Dear Reviewer yKhn**
> >
> > **We sincerely appreciate your detailed and insightful feedback on our paper**. We are grateful for your recognition of our work’s contributions, and in response, we have made several key clarifications to further strengthen the manuscript, including the following:
> >
> > 1. **Brief description of baselines:** We have revised a portion of the Related Work to emphasize our contributions.
> >
> > 2. **Metric definitions:** We have added formal definitions of accuracy and macro-F1 metrics.
> >
> > 3. **Numerical results:** We have provided complete numerical results with standard deviations in tabular format.
> >
> > 4. **Hyperparameter consistency:** We have clarified hyperparameter consistency across datasets with minimal adaptive tuning.
> >
> > **We hope these responses adequately address your comments**. Given the tight timeline, we kindly encourage you to consider our rebuttal when finalizing your evaluation. **Should you have any further comments or suggestions, please do not hesitate to reach out. We remain available throughout the revision process.**
> >
> > Thank you again for your constructive and thoughtful feedback.
> >
> > Best regards,
> >
> > The authors

---

> ### Comment · Reviewer_yKhn · 2025-08-05
> **Concerns addressed, raising score to Accept**
>
> Thank you for the detailed rebuttal and additional analysis. The inclusion of CoUDA as a baseline strengthens the empirical evaluation, and I appreciate the clarification that most hyperparameters remain fixed across datasets, with only one tuned parameter - this improves practical feasibility. The explanation of how the proposed method differs from prior work, particularly through the use of global historical information rather than relying on previous inputs, was also helpful.
>
> Given these clarifications and the strong comparative results across three datasets, I am satisfied that the paper addresses my earlier concerns on clarity, reproducibility, and empirical support.
>
> I am updating my recommendation to Accept (Score -5).

---

> > ### Author Response · Authors · 2025-08-05
> > **Thank you for raising the score**
> >
> > **Thank you for your prompt and positive reply confirming that our responses have fully addressed your concerns, and for raising your rating score.**  We sincerely appreciate your valuable and constructive feedback, which has greatly improved the clarity and quality of our manuscript. As you recommended, we will integrate all clarifications and additional experimental results into the final version. **Thank you once again for your thoughtful suggestions.**

---

### Official Review · Reviewer_EKab · 2025-07-03

**Clarity:** 3
**Significance:** 2
**Originality:** 2
**Rating:** 4
**Confidence:** 3

**Summary:**

This paper presents SPICED, a neuromorphic framework designed for unsupervised continual EEG decoding in dynamic, real-world scenarios where new individuals with distinct EEG patterns are continuously introduced. Inspired by the synaptic homeostasis mechanism in the human brain, SPICED addresses the long-standing stability-plasticity dilemma in continual learning.

By bridging biological principles with neuromorphic continual learning, SPICED opens new directions for bio-inspired unsupervised learning systems, particularly in brain–computer interface (BCI) and non-stationary signal decoding applications.

**Questions:**

1) To what extent are the bio-inspired mechanisms in SPICED necessary for performance, versus engineering design choices that could be replaced with other techniques (e.g., replay buffers, attention-based routing)?

2) How does SPICED handle extremely noisy EEG signals, domain shifts across populations (e.g., clinical vs. healthy), or longer task sequences?

**Ethical Concerns:**

["NO or VERY MINOR ethics concerns only"]

**Final Justification:**

The topic is engaging and of clear interest to the community. While the work presents promising ideas, the current version has notable gaps in completeness and development. Following the rebuttal and discussions, some clarifications were provided. Balancing the novelty of the topic with the present limitations, I recommend a Borderline accept.

**Limitations:**

yes

**Quality:**

3

**Strengths And Weaknesses:**

Quality: The paper presents a well-motivated and carefully implemented framework for continual learning in EEG decoding, grounded in biological principles.

Clarity: The paper is clearly written and well-organized. Key concepts such as critical memory reactivation, consolidation, and renormalization are clearly introduced and motivated through biological analogies.

Weaknesses: The paper does not provide details on the computational overhead or resource requirements of SPICED, particularly in terms of memory expansion and replay mechanisms. Understanding the scalability of the model is crucial for real-time BCI applications.

The paper does not analyze where SPICED may fail or behave suboptimally—e.g., in the presence of low-quality EEG signals, highly non-overlapping user domains, or very long task sequences.

---

> ### Author Rebuttal · Authors · 2025-07-31
>
> **We'd like to express our sincere gratitude for your careful readings and valuable comments.** We provide our feedbacks as follows.
>
> ## Main Comments:
>
> > #### **Q1: The paper does not provide details on the computational overhead or resource requirements of SPICED.**
>
> **A1: Thanks for this constructive comment.** We have added new experiments to evaluate the computational cost of SPICED. Specifically, we report the average time and storage required for SPICED to adapt to each new individual in the continual learning pipeline. As shown in the table below, **the reported average time per individual includes the full adaptation process**: synaptic node incorporation, critical memory reactivation, model training (including synaptic consolidation and weight renormalization) and evaluation. The reported storage refers to the disk storage usage per individual. These results demonstrate that SPICED achieves efficient adaptation with manageable computational overhead. And our model is trained on a single machine equipped with an Intel Core i9 10900K CPU and eight NVIDIA RTX 3080 GPUs.
>
> |     | ISRUC | FACED | Physionet-MI |
> | --- | --- | --- | --- |
> | Average Cost (minutes) | 4.42±0.55 | 4.16±0.64 | 4.23±0.63 |
> | Storage (M) | 47.1 | 53.7 | 43.7 |
>
> ---
>
> > #### **Q2: How does SPICED handle extremely noisy EEG signals, domain shifts across populations (e.g., clinical vs. healthy), or longer task sequences?**
>
> **A2: We thank the reviewer for raising this important point about the robustness of SPICED under extreme conditions.** We address the three extreme scenarios raised regarding the robustness of SPICED as follows:
>
> 1. **Condition of Noisy EEG Signals:** We have added Gaussian noise scaled to 1%, 5%, and 10% of the original signal’s standard deviation during training phase (i.e., progressively noisier conditions). **Due to time constraints, we conducted experiments under the source domain split ratio of 30%**, with each experiment repeated for 3 runs to ensure reliable statistical evaluation. And the results are summarized in the table below. Results are reported for $M_0$​ (initial source model) and $M_i$ (adapted individual model) across all the three datasets.
>
>   | ISRUC | ACC $M_0$ | ACC $M_i$ | MF1 $M_0$ | MF1 $M_i$ |
>   | --- | --- | --- | --- | --- |
>   | 1% Noise | 66.8 | 74.0±0.24 | 60.5 | 70.2±0.21 |
>   | 5% Noise | 66.8 | 74.2±0.08 | 60.5 | 70.4±0.21 |
>   | 10% Noise | 66.8 | 73.8±0.26 | 60.5 | 70.1±0.24 |
>   | Clean EEG | 66.8 | 75.6±0.13 | 60.5 | 71.4±0.07 |
>
>   | FACED | ACC $M_0$ | ACC $M_i$ | MF1 $M_0$ | MF1 $M_i$ |
>   | --- | --- | --- | --- | --- |
>   | 1% Noise | 31.7 | 43.1±0.41 | 27.2 | 40.8±0.40 |
>   | 5% Noise | 31.7 | 43.2±0.29 | 27.2 | 40.9±0.36 |
>   | 10% Noise | 31.7 | 43.0±0.21 | 27.2 | 40.8±0.24 |
>   | Clean EEG | 31.7 | 43.7±0.51 | 27.2 | 41.7±0.58 |
>
>   | Physionet-MI | ACC $M_0$ | ACC $M_i$ | MF1 $M_0$ | MF1 $M_i$ |
>   | --- | --- | --- | --- | --- |
>   | 1% Noise | 42.2 | 48.7±0.14 | 37.9 | 47.8±0.12 |
>   | 5% Noise | 42.2 | 48.6±0.14 | 37.9 | 47.8±0.14 |
>   | 10% Noise | 42.2 | 48.7±0.05 | 37.9 | 47.8±0.08 |
>   | Clean EEG | 42.2 | 48.7±0.17 | 37.9 | 47.9±0.24 |
>
>   As shown in the tables, SPICED exhibits **remarkable robustness** to noise:
>
>   - **On ISRUC**, even under 10% added noise, SPICED​ maintains an ACC of 73.8 and MF1 of 70.1, with only marginal degradation compared to the clean condition (75.6 ACC, 71.4 MF1).
>   - **On FACED**, SPICED shows nearly identical performance across all noise levels, with $M_i$ ACC fluctuating within ±0.7 of the clean setting (43.7 ± 0.51), and MF1 within ±0.9.
>   - **On Physionet-MI**, performance is virtually unaffected across all noise levels, with deviations well within standard error.
>
>   **This suggests that SPICED is inherently robust to noisy EEG signals**, and can effectively handle moderate levels of input perturbations without significant performance degradation.
>
> 2. **Condition of Domain Shifts:** In fact, due to significant inter-subject variability, substantial domain shift can exist even within the same population, **which is particularly evident in outlier subjects**. This is discussed in Section 4.2.1, where Figure 3 shows that each dataset contains outliers whose distributions deviate noticeably from the overall data distribution and exhibit significant domain shift. When encountering a new individual with significant domain shift, **SPICED integrates information across the holistic synaptic network for adaptation**, rather than relying on incremental updates from the previous time step. By leveraging global historical information, SPICED effectively mitigates the limitations of prior methods, which often fail to adapt to outliers exhibiting large domain shifts.
>
> 3. **Condition of Longer Task Sequences:** Regarding dataset selection, we have evaluated SPICED on three **large-scale datasets** composed of at least **100** subjects to validate its performance under long task sequences. While many publicly available EEG datasets consist of only dozens of subjects, our choice of large-scale benchmarks enables a more realistic and rigorous evaluation of long-term individual continual learning. Besides, as discussed in Section 4.2.1, we further analyze SPICED’s behavior under various source/target subject splits. Notably, in the 10% split setting, **the number of individuals in the task sequence is nine times the number of pretraining individuals.** For instance, in FACED, only **12** subjects are assigned to the pretraining set, while the remaining **111**subjects form the long sequential stream for continual learning. We believe this constitutes a sufficiently long and challenging continual learning trajectory, and thus provides strong evidence for SPICED’s effectiveness in long-term continual learning scenarios.
>
>
> **We hope that these points will address your concerns.** Thank you once again for your insightful feedback.
>
> ---
>
> > #### **Q3: To what extent are the bio-inspired mechanisms in SPICED necessary for performance, versus engineering design choices that could be replaced with other techniques (e.g., replay buffers, attention-based routing)?**
>
> **A3: Thank you for this insightful and constructive question.** We clarify SPICED’s necessity through its core innovation and its implications for prior approaches:
>
> 1. **Innovations of SPICED:** Our framework introduces an auxiliary synaptic network that **aggregates global historical information** (across all prior individuals) for decision-making when adapting to new tasks and activates **the most relevant memories**(e.g., critical memory replay, critical model fusion) to assist adaptation. The selection of critical memory is enabled by the following two operations:
>
>   - Task-discriminative memory strengthening via **synaptic consolidation**, and
>   - Redundant noisy memory weakening via **synaptic renormalization**.
> 2. **Limitations of Traditional Techniques:** The proposed replay buffers or attention-based routing methods necessarily **depend on the model’s immediate prior state**. When significant domain shift occurs between tasks (e.g., due to inter-subject EEG variability), these methods lose access to globally relevant historical knowledge—making them sensitive to the learning trajectory and prone to performance degradation. Crucially, without global memory aggregation, existing methods (e.g., replay buffer-based approaches) **cannot selectively retrieve and replay the most task-discriminative memories** from arbitrary points in the past.
>
> 3. **Resolution via SPICED:** We translate the synaptically inspired mechanism into a continual learning framework, overcoming the aforementioned limitations. Our synaptic network:
>
>   - Maintains a consolidated memory pool spanning all historical subjects
>   - Dynamically reactivates only task-critical memories during adaptation
>   - Ensures robustness to continual domain shifts
>
> To sum up, SPICED is not replaceable by conventional techniques—it solves a fundamental limitation in continual learning under cross-subject domain shifts.
>
> **We appreciate your insightful question and hope this clarifies the necessity of SPICED’s bio-inspired design compared to purely engineering-driven alternatives.**

---

> > ### Author Response · Authors · 2025-08-05
> > **Rebuttal has been submitted and we are eager to hear your further constructive feedback**
> >
> > **Dear Reviewer EKab**
> >
> > We hope this message finds you well. We sincerely appreciate your valuable feedback on our paper. In response, we have made substantial clarifications to address your concerns, including the following:
> >
> > 1. **Computational Efficiency:** We have added new experimental results quantifying the computational cost of SPICED.
> >
> > 2. **Robustness to Extreme Conditions:** We have expanded our discussions and clarifications to three challenging scenarios:
> >
> >     (1) **Condition of Noisy EEG Signals**
> >
> >     (2) **Condition of Domain Shifts**
> >
> >     (3) **Condition of Longer Task Sequences**
> >
> > 3. **Necessity of Bio-Inspired Design**:  We clarify SPICED’s necessity through its core innovation and its implications for prior approaches.
> >
> > **We hope these responses provide clear and convincing answers to your insightful questions.** Given the tight timeline of the review process, we kindly encourage you to consider our rebuttal when finalizing your evaluation. Should you have any further comments or suggestions, please do not hesitate to reach out. We remain available throughout the revision process.
> >
> > Thank you again for your constructive and thoughtful feedback.
> >
> > Best regards,
> >
> > The authors

---

> > ### Comment · Reviewer_EKab · 2025-08-06
> >
> > Thank you for your detailed responses. I have carefully read them and believe they address most of my concerns. Therefore, I will keep my score as 4: Borderline Accept.

---

> > > ### Author Response · Authors · 2025-08-06
> > > **Thanks for your timely reply and remaining postive rating**
> > >
> > > **We greatly appreciate your careful readings and insightful comments, which have helped us improve the clarity and rigor of the paper. We are pleased to see that our responses have addressed your concerns.**
> > >
> > > Should you have any further comments or suggestions, please do not hesitate to reach out. We remain available throughout the revision process. **Thank you once again for your valuable insights.**

---

### Note · Authors · 2025-08-11

**We first sincerely appreciate the NeurIPS review process, which has enabled valuable exchanges with reviewers of diverse perspectives, significantly improving our paper and contributing to the EEG community.**

**And we thank the reviewers for their insightful feedback.** We appreciate the positive comments on the the biological motivation of SPICED (Ekab, yKhn, zYsa), the impact on real-world scenarios (Ekab, yKhn), the clear presentation and organization (Ekab, yKhn, HgnD), the different evaluated datasets (yKhn, zYsa, HgnD), the superior results (yKhn, HgnD), the clear diagrams and visualizations (yKhn, zYsa).

We acknowledge some reviewers' concerns regarding the detailed analysis on memory cost, the detailed dataset partition, some missing related works, the detailed discussion of extreme cases, more neuroscience literature for biological claims, more compared baselines, the choice of evaluation metrics. **During this rebuttal, we have addressed all the reviewers’ concerns through additional experiments, detailed analysis and explanations, and thorough discussion.** This represents the best effort we can achieve within the limited timeframe allocated for the rebuttal.

**We are grateful to reviewers Ekab and yKhn for their recognition of our work, and are pleased that our responses have adequately addressed their concerns.**

**We also thank reviewers zYsa and Hgnd for their constructive feedback.** Due to the lack of interaction during the discussion phase, we are uncertain of their final assessments.

**For the reviewer zYsa**, we provided detailed responses on:

1. the unsupervised claim,

2. continual learning metrics

3. biological plausibility

4. low-resource performance

5. generalizability of SPICED

6. baseline comparisons.


**For the reviewer Hgnd**, we clarified the distinctions between SPICED and classical GNG/SOM in terms of

1. **biological motivation**

2. **functional mechanism**

3. **application objective**

 And we **corrected a misunderstanding arising from an incorrect interpretation of the data split.**

**We believe our responses fully address the reviewers’ questions, and we are committed to incorporating their feedback to further strengthen the work.**

---

### Decision · Program_Chairs · 2025-09-17

**Decision:**

Accept (poster)

**Comment:**

This paper presents SPICED, a synaptic homeostasis-inspired framework for unsupervised continual EEG decoding. The biological motivation is clear, and the work is well-written and organized, with thoughtful integration of consolidation, reactivation, and renormalization mechanisms. Evaluation across three EEG datasets shows consistent improvements, and the rebuttal provided additional clarity, including computational cost analysis and robustness tests. However, the improvements over prior baselines are modest and were originally reported only in bar plots without numerical detail, limiting interpretability. More importantly, compute overhead comparisons against prior methods remain incomplete, leaving questions on practical scalability. Overall, while the paper is not without weaknesses, the novelty, biological grounding, and clarified empirical support justify a poster acceptance.

**Authors are encouraged to provide explicit comparisons of computational overhead (e.g., runtime, memory, FLOPs) with prior methods, as this would significantly strengthen the empirical evaluation and highlight the practical scalability of the proposed framework.**